# An efficient and stable solar flow battery enabled by a single-junction GaAs photoelectrode

Hui-Chun Fu[1,2,5], Wenjie Li[1,5], Ying Yang[1,3,5], Chun-Ho Lin[2], Atilla Veyssal[1], Jr-Hau He[4✉] & Song Jin[1✉]

Converting and storing solar energy and releasing it on demand by using solar flow batteries (SFBs) is a promising way to address the challenge of solar intermittency. Although high solar-to-output electricity efficiencies (SOEE) have been recently demonstrated in SFBs, the complex multi-junction photoelectrodes used are not desirable for practical applications. Here, we report an efficient and stable integrated SFB built with back-illuminated single-junction GaAs photoelectrode with an n-p-n sandwiched design. Rational potential matching simulation and operating condition optimization of this GaAs SFB lead to a record SOEE of 15.4% among single-junction SFB devices. Furthermore, the $TiO_2$ protection layer and robust redox couples in neutral pH electrolyte enable the SFB to achieve stable cycling over 408 h (150 cycles). These results advance the utilization of more practical solar cells with higher photocurrent densities but lower photovoltages for high performance SFBs and pave the way for developing practical and efficient SFBs.

[1] Department of Chemistry, University of Wisconsin-Madison, 1101 University Avenue, Madison, WI 53706, USA. [2] Division of Computer, Electrical and Mathematical Sciences and Engineering, King Abdullah University of Science and Technology, Thuwal 23955-6900, Saudi Arabia. [3] Shaanxi Provincial Key Laboratory of Electroanalytical Chemistry, Key Laboratory of Synthetic and Natural Functional Molecule Chemistry of the Ministry of Education, College of Chemistry & Materials Science, Northwest University, Xi'an 710127, China. [4] Department of Materials Science and Engineering, City University of Hong Kong, Kowloon, Hong Kong, China. [5]These authors contributed equally: Hui-Chun Fu, Wenjie Li, Ying Yang. ✉email: jrhauhe@cityu.edu.hk; jin@chem.wisc.edu

The increasing demand for clean and renewable energy has stimulated the development of many important technologies for simultaneous conversion and storage of intermittent solar energy[1–4]. Among them, solar-driven photoelectrochemical (PEC) water splitting to produce hydrogen[5–7] and reduction of carbon dioxide for the production of fuels[8,9] have received considerable research interests due to the promises for harvesting solar energy and storing it as chemical energy[10,11]. However, practical PEC technologies are impeded by several key challenges[2,10,12,13], including the sluggish kinetics of fuel-producing reactions that require efficient and robust electrocatalysts, the poor stability of many photoelectrode materials under PEC conditions, and the need for additional fuel cell devices to regenerate electricity from solar fuels. Recently, solar flow batteries (SFBs)[14–18] that monolithically integrate photovoltaics (PVs) or regenerative PEC cells and redox flow batteries (RFBs)[19,20] have emerged as an alternative approach to avoid the aforementioned issues of conventional PEC devices yet achieve the same function of harvesting and storing solar energy into chemical energy. In SFBs, redox couples with facile kinetics are used to store and release solar energy as electricity under mild electrochemical conditions. This eliminates the need for electrocatalysts and separate fuel cell devices and could relax the stability requirement for photoelectrode materials.

In order to achieve high-performance SFBs, efficient solar cells[21–25] are needed. III–V semiconductor materials are commonly used for high-efficiency PV applications due to their direct bandgap, high absorptivity for sunlight, high electron mobility, and well-controlled crystal growth[21,26,27]. The integration of a triple-junction III–V (InGaP/GaAs/Ge) photoelectrode with aqueous organic 4-hydroxy-2,2,6,6-tetramethylpiperidin-1-oxyl (4-OH-TEMPO) and methyl viologen redox couples in an SFB device yielded a round-trip solar-to-output electricity efficiency (SOEE) of 14.1% for SFBs[15]. Note that the SOEE of an integrated SFB device is calculated using the following equation[14]:

$$\text{SOEE} = \frac{E_{\text{electrical,out}}}{E_{\text{illumination}}} = \frac{\int I_{\text{out}} V_{\text{out}} dt}{\int SA dt}, \qquad (1)$$

where the $E_{\text{illumination}}$ is the incident solar energy, $E_{\text{electrical,out}}$ is the output electrical energy from discharging the SFB, $I_{\text{out}}$ and $V_{\text{out}}$ are the output current and voltage during discharge, respectively, $S$ is the total incident solar irradiance, and $A$ is the area of the light-harvesting window of the photoelectrode. Despite the high SOEE and the conceptual advance, this III–V based SFB suffers from several shortcomings. First, the maximum power point voltage ($V_{\text{MPP}}$) of the tandem photoelectrode is much higher than the cell potential of SFB ($E^0_{\text{cell}}$ which is determined by the formal potential difference between the anolyte and catholyte redox couples, i.e., $E^0_{\text{cell}} = E^0_{\text{anolyte}} - E^0_{\text{catholyte}}$), therefore, a large portion of the high photovoltage is not utilized[15]. Second, the fabrication cost of triple-junction III–V solar cells is too high for practical applications. Third, the Ge bottom cell in the photoelectrode is prone to photocorrosion in aqueous solutions, which has limited the lifetime of the SFB device[15].

On the other hand, GaAs ($E_g = 1.42$ eV) solar cells are the most likely to reach the Shockley–Queisser limit[22,28,29] due to its optimal bandgap and hold the record power conversion efficiency (PCE) of 29.1% for single-junction (SJ) solar cells[21]. In addition, the open-circuit voltage ($V_{\text{OC}}$) of SJ-GaAs cells, usually between 0.9–1.1 V[22], is within the optimal voltage matching range for many aqueous RFBs. Even though GaAs solar cells are still more expensive than silicon solar cells, these attributes make SJ-GaAs photoelectrodes potentially promising for high-performance SFBs, yet they have not been exploited for SFBs. The common GaAs PV cells[30], as well as (tandem) PEC photoelectrodes based on III–V materials[31,32], often adopt an n (emitter)-on-p (base) device structure, which has better carrier collection efficiency[30,33],

using p-type GaAs substrates. However, n-type GaAs substrates are less costly due to the easier fabrication proces[34,35]. Furthermore, it is desirable to harvest photons at the epitaxial n–p junction side, but immerse the protected substrate side of the photoelectrodes in contact with electrolytes for electrochemical reactions[6,32]. These design constraints mean that, in order to enable efficient, stable, and practical SFB devices, we need to design unconventional back-illuminated SJ GaAs solar cells based on n-type GaAs substrates to achieve high carrier collection efficiency and reduced production cost at the same time.

In this work, we present an efficient and stable SFB based on a back-illuminated SJ-GaAs photoanodes with an unusual n–p–n sandwich structure using n-GaAs substrates that are integrated with robust bis ((3-trimethylammonio)propyl)-ferrocene dichloride (BTMAP-Fc), bis (3-trimethylammonio)propyl violo-gen tetrachloride (BTMAP-Vi), and N-methyl-2,2,6,6-tetra-methylpiperidin-1-oxyl ($N^{\text{Me}}$-TEMPO) redox couples in neutral pH electrolytes. To optimize the SOEE of the SFB, we carried out numerical simulations to find the relationship between SOEE and the operating state-of-charge (SOC) range. The highly efficient SJ-GaAs photoelectrode, good potential match, and rational operating condition engineering led to an average SOEE of 13.3% by using the BTMAP-Vi/BTMAP-Fc redox couples. Furthermore, the robust neutral pH aqueous redox couples used and the $TiO_2$ thin film protecting the GaAs substrate from photocorrosion significantly extended the lifetime of the III–V photoelectrode based SFB to more than 400 h (over 150 cycles). Based on a more refined voltage matching analysis of the actual performance of the SJ-GaAs photoelectrode, we further developed another SFB device with better matched BTMAP-Fc and $N^{\text{Me}}$-TEMPO redox couples and demonstrated an average SOEE of 15.4%. This work further reveals insights on how to effectively utilize solar cells that have higher photocurrent densities but lower photovoltages for high-performance SFBs toward practical applications.

## Results

**Design of the SJ-GaAs solar cell.** In order to achieve a good operating potential match between the photoelectrode and aqueous redox couples, we first fabricated and investigated the SJ-GaAs solar cells with an unusual "reversed" n–p–n sandwiched layer stacking with cost-effective n-GaAs substrates. As illustrated in Fig. 1a from bottom to top, the SJ-GaAs solar cell was fabricated by growing a p-on-n tunnel diode on an n-type GaAs substrate followed by an n (emitter)-on-p (base) active junction (see "Methods" for complete fabrication details) We use n-type GaAs substrates for the photoelectrodes due to several advantages over the p-type substrates that are commonly used commercially: n-type GaAs has a lower surface recombination velocity than that of p-GaAs[36] and n-GaAs substrates are more affordable due to fabrication challenges in the p-type doping process[35]. However, a higher carrier collection efficiency can be realized in GaAs solar cells with the epitaxial growth of n (emitter)-on-p (base), instead of p (emitter)-on-n (base), because the electron diffusion length of the p-type base is much longer than the hole diffusion length of the n-type base[30,33]. In order to achieve both better device efficiency and lower cost, here we use an unusual "reversed" layer architecture with an n–p–n sandwich geometry, i.e., both the light-harvesting window and photoelectrode/ electrolyte interface using n-type GaAs. Furthermore, we need a tunnel diode (Te doped-GaAs/C doped-AlGaAs) between the n-type substrate and the p–n junction to act as an Ohmic resistor with the purpose of minimizing dopant diffusion[37,38].

Compared with multijunction III–V photoelectrodes that exhibit high photovoltages (>2.0 V), the SJ-GaAs photoelectrodes feature lower photovoltages (0.9–1.1 V) but higher photocurrents (>21 mA cm$^{-2}$)[21,22,39]. The current density–voltage ($J$–$V$)

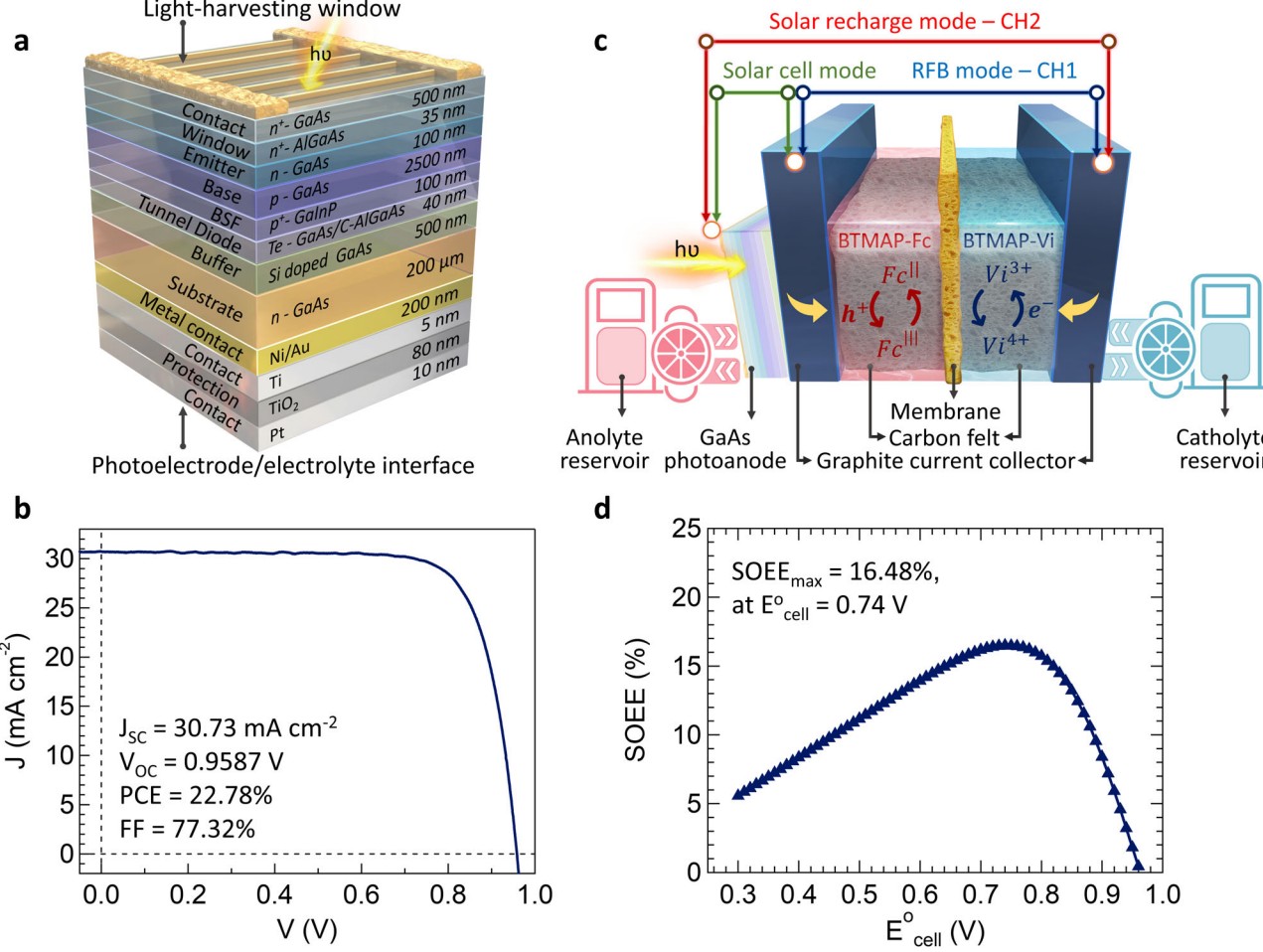

**Fig. 1 Schematic design of the SJ-GaAs solar cell and the SFB device and its PV performance. a** The cross-section schematic and layer information of the SJ-GaAs solar cell design. The contact/protection (Ti/TiO₂) layer at the bottom of the device stack will be in direct contact with the aqueous electrolyte and light illumination comes from the top n⁺-GaAs contact layer. **b** J–V performance of the solid-state SJ-GaAs cell. **c** Schematic illustration of the integrated SFB device, showing a GaAs photoanode (as shown in panel (**a**)) on the left side and two carbon felts inert in two electrolyte compartments separated by a Selemion DSV anion exchange membrane. Illumination comes from the left side to the GaAs photoelectrode. The SFB device can be configured to three different operation modes: solar cell (green), RFB (blue) monitored by Channel 1 (CH1), and solar recharge (red) mode monitored by channel 2 (CH2). **d** Numerically calculated SOEE as a function of $E^0_{cell}$ by using J–V data from (**b**).

performance of the solid-state SJ-GaAs solar cell was then evaluated under one sun (100 mW cm⁻²) of AM 1.5 G illumination (see "Methods"), which revealed a short-circuit current density ($J_{SC}$) of 30.73 mA cm⁻², a $V_{OC}$ of 0.958 V, a fill-factor (FF) of 77.32% and a resulting PCE of 22.78% (Fig. 1b). Note that, a protection layer of Ti/TiO₂/Pt was deposited on the electrolyte contacting surface of the photoelectrode as illustrated in Fig.1a. The J–V performance of the GaAs cells with and without Ti/TiO₂/Pt layer is shown in Supplementary Fig. 1. The DC series resistance ($R_{DC}$) of the GaAs cell calculated according to the J–V curve showed a less than 0.3% increase after depositing the metal oxide layer. We further characterized the absorbance spectra and the external quantum efficiency (EQE, which is defined as the ratio between the externally collected photo-generated carriers of the solid-state solar cell to the incident photons, details in the Methods and Supplementary Fig. 2) of the GaAs cell. These results reveal the outstanding optical and electrical performances of the GaAs cell and maximum absorption in the desired solar spectrum range. By integrating the product of the EQE spectral response and the photon flux of the AM 1.5 G solar spectrum, the corresponding converted current density of the GaAs cell can be calculated to be 26.65 mA cm⁻², in reasonable agreement with

the experimentally measured $J_{SC}$ (30.73 mA cm⁻²). Note that the difference between the converted current density and $J_{SC}$ is likely caused by the different light sources used in the J–V (uniform illumination) and EQE measurements (point-like illumination).

**Design of the SJ-GaAs SFB.** Figure 1c illustrates the design of the integrated SFB device, consisting of a back-illuminated SJ-GaAs photoanode, two carbon felt electrodes, and anolyte/catholyte separated by a Selemion DSV anion exchange membrane (see "Methods"). Such design allows us to switch the function of the device among 3 different modes: solar cell mode (green line in Fig. 1c), for direct electrical energy delivery without storage; RFB mode (blue line), for operating the device as a normal RFB; and solar recharge mode (red line), for converting solar energy to charge up the redox couples. Our unusual design of n–p–n sandwiched SJ-GaAs photoanode enables a back-illuminated design that allows for light absorption from the top n⁺-GaAs side (the light-harvesting window in Fig. 1a, see photograph in Supplementary Fig. 3a), and photoexcited hole extraction from the n-GaAs substrate side (the photoelectrode/electrolyte interface). The GaAs substrate is further protected by a TiO₂ protection layer (80 nm), then in contact with the anolyte through a thin conductive metal layer for the charging

process. The GaAs photoelectrode was sealed onto a custom-made graphite plate, which has an open window of 9 mm × 9 mm at the center (see photograph in Supplementary Fig. 3b, and fabrication details in the Methods section) and assembled into the SFB device to expose the carrier transfer surface of GaAs. Such back-illumination design is uncommon among previously reported III–V semiconductor-based PEC devices[40–44] and provides more freedom to utilize thick and/or opaque protection layers without affecting light-harvesting capability of the photoelectrode.

The stability of the SFB photoelectrode in direct contact with aqueous electrolyte is also critical. Although the chemical and PEC stability of GaAs substrates is better in comparison with that of Ge used in the previous III–V tandem solar cells[15], it is still far from desired for practical long term operation. The stability of GaAs in aqueous electrolytes with different pH also varies significantly with the poorest stability in solutions with extreme pH values (higher than pH 12 or lower than pH 3) and the best stability in neutral solutions[45]. Previous reports have shown that the introducing of a TiO$_2$ surface protection layer on III–V semiconductors can effectively protect the photoelectrodes from photocorrosion[32,40,42,46]. Therefore, we further deposited a TiO$_2$ thin film by atomic layer deposition (ALD) on the electrolyte contacting surface side of GaAs cells (Fig. 1a) to serve as the protection layer and enable stable long term operation. A thin layer of Ti (5 nm) was also deposited before the TiO$_2$ layer of 80 nm to promote adhesion and another thin layer of Pt (10 nm) was deposited on TiO$_2$ to enhance charge extraction at the photoelectrode/electrolyte interface.

**Potential match modeling of SFB and performance estimate**. In order to achieve an efficient and stable SFB, we need to match the $E^0_{cell}$ of the SFB with the photovoltage of the photoelectrodes[14] and satisfy the essential requirements for robust and noncorrosive electrolytes. To find the best-matched $E^0_{cell}$ for the SJ-GaAs solar cell, we used a numerical modeling method to simulate the relationship between SOEE and $E^0_{cell}$ using the $J$–$V$ curve of the solid-state GaAs solar cell (shown in Fig. 1b, see "Methods" for simulation details). As a result, the SOEE-$E^0_{cell}$ simulation predicts an optimal $E^0_{cell}$ of 0.74 V with a maximum SOEE (SOEE$_{max}$) of 16.5% (Fig. 1d). To find stable redox couples with good potential matches with the SJ-GaAs cell, we turn to the neutral pH BTMAP-Vi and BTMAP-Fc redox couples. Enabled by the strong electrostatic repulsion from the positively charged BTMAP side chains and the relatively large molecular size of the two BTMAP redox couples, the RFB built with these redox couples is one of the most stable neutral pH aqueous organic RFBs reported so far[47]. More importantly, the $E^0_{cell}$ matches the optimized voltage predicted above well for the SJ-GaAs cell. The electrochemical properties of the BTMAP-Vi and BTMAP-Fc redox couples were characterized by three-electrode cyclic voltammetry (CV) measurement. As shown in Fig. 2a, the formal potential of BTMAP-Vi $\left(E^0_{Vi}\right)$ and BTMAP-Fc $\left(E^0_{Fc}\right)$ are −0.353 and 0.382 V vs. SHE, respectively. Note that the SJ-GaAs cell acts as a photoanode in this work, while a graphite plate is at the cathode side for charging BTMAP-Vi. Although the formal potential of BTMAP-Vi is −0.353 V vs. SHE, the overpotential of the graphite plate is too high for the hydrogen generation reaction to occur. Hence, an $E^0_{cell}$ of 0.735 V for the SFB could be estimated. The charge/discharge reactions of these two redox couples are described below:

$$Vi^{4+} + e^- \rightarrow Vi^{3+} \text{(charge)} \text{and} Vi^{3+} - e^- \rightarrow Vi^{4+} \text{(discharge)}; E^0_{Vi} = -0.353 \text{ V}.$$
(2)

$$Fc^{II} - e^- \rightarrow Fc^{III} \text{(charge)} \text{and} Fc^{III} + e^- \rightarrow Fc^{II} \text{(discharge)}; E^0_{Fc} = 0.382 \text{ V}.$$
(3)

Excellent stability was demonstrated for a neutral pH RFB built with 0.20 M BTMAP-Vi/BTMAP-Fc redox couples and cycled galvanostatically at different current densities of 5–100 mA cm$^{-2}$ for 5 cycles each (Supplementary Fig. 4). Figure 2b displays the Coulombic efficiency (CE, green dots), voltage efficiency (VE, red dots) and energy efficiency (EE, blue circles) according to the RFB galvanostatic cycling results. The equations for calculating the CE, VE, and EE are in the Methods. The RFB maintained a nearly constant CE > 99% over 29 h. The cell potential-capacity profile during the galvanostatic cycling test of the RFB with 0.20 M BTMAP-Vi/BTMAP-Fc redox couples revealed an average galvanostatic–potentiostatic charge/discharge capacity of 2.41 Ah L$^{-1}$ (energy density of 2.05 Wh L$^{-1}$) that could be obtained even at a high current density of 50 mA cm$^{-2}$ (Supplementary Fig. 5). This is very close to the theoretical capacity of 2.68 Ah L$^{-1}$. Due to the excellent cycling stability and good voltage match for the SJ-GaAs photoelectrode, we built up an integrated SJ-GaAs SFB device with these neutral pH BTMAP-Vi/BTMAP-Fc redox couples and evaluated its overall charge/discharge performances.

**Optimization of SFB operating conditions**. Because SJ GaAs solar cells yield a high photocurrent density of about 30 mA cm$^{-2}$, we first need to optimize the electrolyte concentration and flow rate in the integrated SFB device to prevent the accumulation of the photoexcited carriers at the photoelectrode/electrolyte interface. We carried out linear sweep voltammetry (LSV) measurements of the integrated SFB device with different redox couple concentrations of 0.1 and 0.2 M, and various flow rates from 20 to 120 mL min$^{-1}$ (measured under solar cell mode, as shown in Supplementary Fig. 6). We observed improvements of the $J_{sc}$, $V_{oc}$, and FF with the higher redox couple concentration and flow rates, which indicate more facile electrode kinetics and charge carrier transport[48]. These measurements showed that a concentration of 0.1 M results in much lower $J_{sc}$ but a concentration of 0.2 M and an optimized flow rate of 60 mL min$^{-1}$ can support sufficient electrochemical mass transport between the SJ-GaAs photoanode and the inert carbon felt. In preliminary SFB cycling tests using the electrolyte flow rates of 40, 60, and 80 mL min$^{-1}$ (10 cycles for each condition as displayed in Supplementary Fig. 7), the overall SFB performance shows a slightly improved with the flow rate higher than 60 mL min$^{-1}$, and the highest initially SOEE of 15.1% can be achieved with the flow rate of 80 mL min$^{-1}$. However, it is at the cost of extremely fast performance decay in only three cycles (Supplementary Fig. 7c, d) due to serious surface destruction. Post-mortem optical imaging (Supplementary Fig. 3c–e) and X-ray photoelectron spectroscopy surface analysis (Supplementary Fig. 8) were further performed to understand the failure mechanism of the surface protection layer of Ti/TiO$_2$/Pt. They suggested that the TiO$_2$ protection layer (together with the Pt layer on top) disappeared after cycling for the case of the 80 mL min$^{-1}$ flow rate, but remained mostly intact for the cases of lower flow rates. The disappeared Pt XPS signal and the newly emerged Ga and As signals after cycling at the 80 mL min$^{-1}$ flow rate indicated that the Ti/TiO$_2$/Pt protection layer was peeled off during the SFB cycling test to expose the vulnerable GaAs substrate to the electrolyte. Therefore, the observed SFB device performance decay was likely caused by such mechanical damage under the high flow rate. Therefore, in the remaining SFB measurements of this work, the electrolyte flow rate was set to 60 mL min$^{-1}$ to balance the efficiency and stability concerns.

Figure 3a shows the LSV behavior of the integrated SFB measured at 50% SOC under solar cell mode with the optimized electrolyte concentration (0.2 M) and flow rate (60 mL min$^{-1}$) and one Sun of AM 1.5 G illumination. The SJ-GaAs photoanode in the integrated SFB device exhibited a $J_{sc}$ of 28.6 mA cm$^{-2}$ (the area of

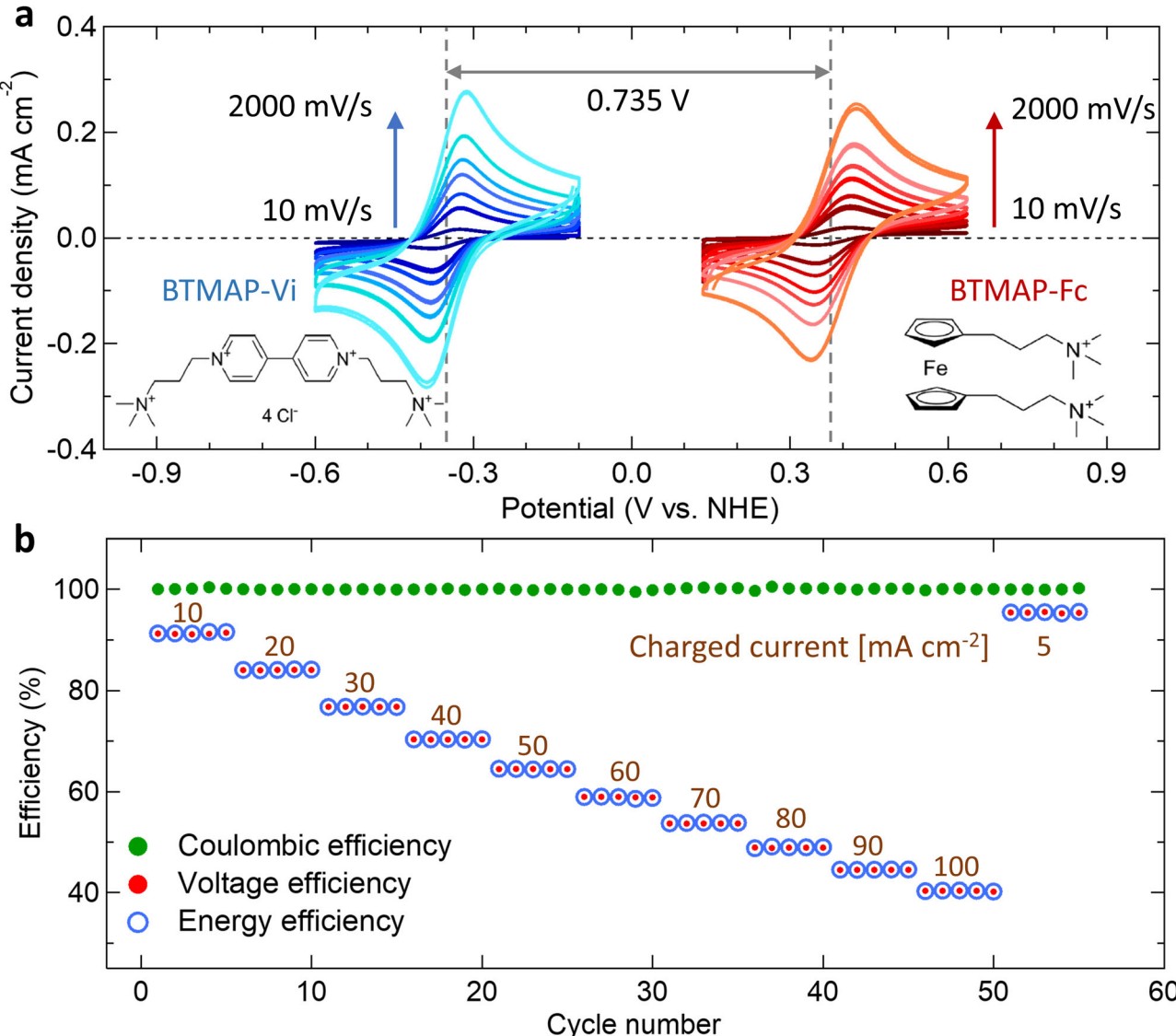

**Fig. 2 CV and RFB cycling performance of BTMAP-Vi and BTMAP-Fc. a** CV of 5 mM BTMAP-Vi (blue curve) and 5 mM BTMAP-Fc (red curve) at the scan rates of 10, 100, 200, 400, 600, 1000, and 2000 mV s$^{-1}$ (color from dark to light). **b** Coulombic efficiency (green dots), voltage efficiency (red dots) and energy efficiency (blue circles) of the RFB cycled galvanostatically with 0.20 M BTMAP-Vi and 0.20 M BTMAP-Fc at different current densities of 5–100 mA cm$^{-2}$ and cut-off voltages of 1.1 and 0.3 V. 1.0 M NaCl was used as the supporting electrolyte for both redox couples in the CV and RFB cycling tests.

the light-harvesting window was 0.53 cm$^2$. See Methods for complete details of the area calculation) and a $V_{oc}$ of 0.936 V, which is in good agreement with the $J$–$V$ performance of the solid-state GaAs cells (Fig. 1b). Note that photocurrent instead of photocurrent density is plotted in Fig. 3a for matching with the $I$–$V$ curves of the RFB. By overlaying a series of $I$–$V$ curves of the integrated SFB device measured under solar cell mode at 50% SOC (red line) and RFB mode at different SOCs (the blue lines from light to dark), the photocurrent provided by the SJ-GaAs photoanode ($I_{operating}$(SOC)) and the corresponding cell potential ($V_{operating}$(SOC)) at various SOCs can be predicted according to their intersection points. The qualitative prediction reveals that the charging photocurrent will decrease dramatically after the integrated SFB was charged over 75% SOC, because of the increasing $E_{cell}$.

The solar conversion efficiency of the integrated SFB will decay with increasing SOC, which can be quantitatively assessed by

calculating the instantaneous SOEE (SOEE$_{ins}$) as the function of SOCs[14]:

$$\text{SOEE}_{\text{ins}}(\text{SOC}) = \frac{P_{\text{electrical,out}}}{P_{\text{illumination}}} = \frac{I_{\text{operating}}(\text{SOC}) \times V_{\text{operating}}(\text{SOC})}{SA} \times \text{CE} \times \text{VE},$$

(4)

where the $P_{\text{electrical,out}}$ and $P_{\text{illumination}}$ are the power of output electricity and incident light, the $I_{\text{operating}}$(SOC) is photocurrent provided by the SJ-GaAs photoanode at various SOCs, $V_{\text{operating}}$(SOC) is the corresponding cell potential, $S$ is the total incident solar irradiance (100 mW cm$^{-2}$), and $A$ is the area of the light-harvesting window of the photoelectrode (see "Methods" for complete details of the area calculation). According to Eq. 4 and the detailed modeling method described in the "Methods" section, the SOEE$_{ins}$-SOC (blue curve) and SOEE$_{ins}$-$E_{cell}$ (red curve) relationships were numerically

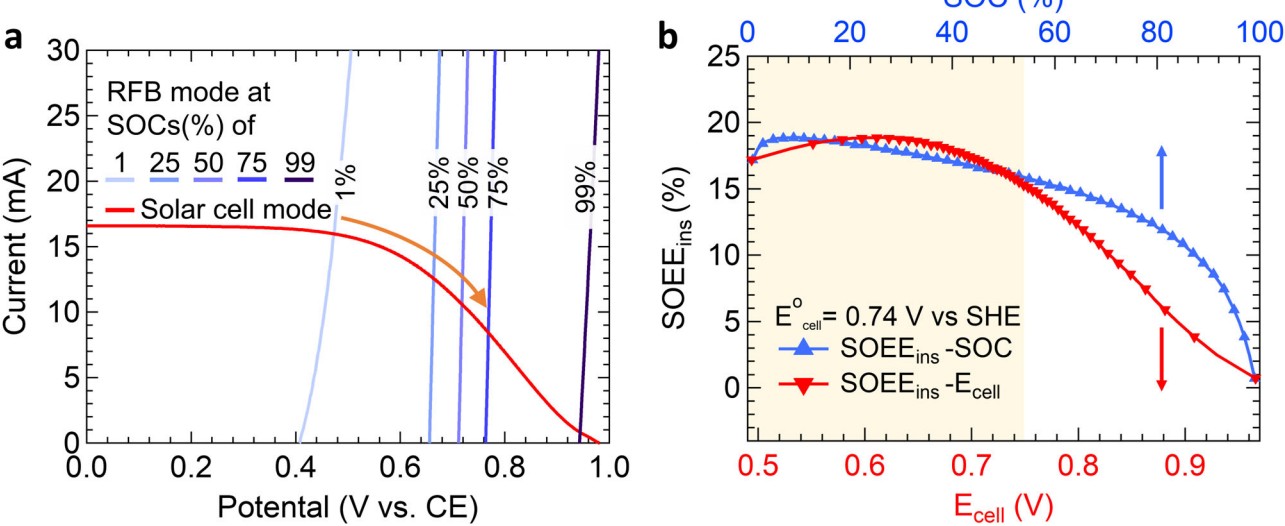

**Fig. 3 Estimation of SOEE_ins for SFB built with SJ-GaAs photoanode and BTMAP redox couples. a** *I–V* performance of the GaAs photoanode at 50% SOC under solar cell mode (red line) and RFB mode (blue lines) in 0.2 M BTMAP electrolytes with the flow rate of 60 mL min$^{-1}$ measured under one Sun illumination. The blue lines from light to dark represent the $E_{cell}$ at 1–99% SOC of the RFB. Note that photocurrent instead of photocurrent density is plotted here. **b** Potential match profile of the SFB's SOEE_ins as a function of $E_{cell}$, (red curve) and the state of charge (SOC, blue curve).

simulated (based on the optimal $E^0_{cell}$ of 0.74 V vs. SHE in Fig. 1d) and shown in Fig. 3b. The SJ-GaAs SFB device can be effectively charged with a high SOEE_ins > 15% at a SOC range from 0 to 54% (the charged capacity of 1.30 Ah L$^{-1}$ when the SFB was charged to 54%), corresponding to a cell potential range from 0.49 to 0.75 V (the yellow shaded area in Fig. 3b).

**Operation and characterization of integrated SFB device.** In light of the comprehensive operating condition analysis of the SJ-GaAs SFB, we performed long term SFB cycling test with the optimized operating conditions (charging the SFB at SOC range from 0 to 54% to ensure the SOEE_ins > 15%). During the cycling test, the integrated SFB was charged under solar recharge mode with simulated one Sun solar illumination and a 1.35 h time limit to control the SOC utilization range (ca. 0-54%), followed by galvanostatic discharging under the RFB mode with a current of 11 mA and a cutoff potential of 0.3 V. We used a synchronized potentiostat with two separated channels to monitor the $E_{cell}$ of the integrated SFB (the blue connection in Fig. 1c and the data are displayed as the blue curve in Fig. 4a) and the charging photocurrent of the SJ-GaAs photoanode (the red connection in Fig. 1c and the data are displayed as the red curve in Fig. 4a) throughout the cycling test. During each charging cycle, the SJ-GaAs photoanode of the SFB showed an initial photocurrent density of ~25.9 mA cm$^{-2}$, which gradually decreased with the increasing SOC as predicted in Fig. 3a, and yielded an average photocurrent density of 11 mA cm$^{-2}$ (i.e., a current of 6.18 mA with the light-harvesting area of the SJ-GaAs photoanode of 0.53 cm$^2$).

This integrated SJ-GaAs SFB reached an impressive initial SOEE of 14.3%. Enabled by the robust BTMAP-Vi/BTMAP-Fc redox couples and the TiO$_2$ protection layer on GaAs photoelectrode, the integrated SFB device was continuously cycled for 150 cycles and showed fairly stable performance with average CE and VE of 98.6% and 96.2%, respectively (Fig. 4b). Over the 150 solar charging and discharging cycles (408 h), the SOEE decreased slightly from the initial value by 5.6%, resulting in an average SOEE of 13.3%. This decay can be attributed to the increased $R_{DC}$ of the SFB (Supplementary Fig. 9a, b), due to the capacity fade (Supplementary Fig. 9c, d) and the surface corrosion of the SJ-

GaAs photoanode over the operation period (Supplementary Fig. 3f). The XPS analysis of the SJ-GaAs photoanode surface after 150 charge/discharge cycles (Supplementary Fig. 10) revealed diminished Pt and Ti signals and newly emerged Ga and As signals, which suggested that the Ti/TiO$_2$/Pt protection layer was significantly damaged or peeled off during the long operation period to expose the vulnerable GaAs substrate to the electrolyte.

In addition to the high SOEE, the solar power conversion utilization ratio (SPUR), defined as the ratio between the SOEE of the SFB and the PCE of the solid-state solar cell, is another important figure of merit for SFBs. Due to the rational $E^0_{cell}$ matching and operating condition optimization, this SJ-GaAs SFB achieved a SPUR of 58.4% (based on a PCE of 22.78% for the SJ-GaAs solar cell). However, there is still considerable room for further improvements. First, the LSV of the GaAs photoanode exhibits a significantly decreased FF (52.77%, Fig. 3a) in comparison with that of the solid-state GaAs solar cell (FF of 77.32%, Fig. 1b). This FF decrease is caused by insufficient charge transfer at the photoanode/electrolyte interface, which is a common issue for photoelectrodes with high photocurrent density (>20 mA cm$^{-2}$), as discussed in previous reports[14,49]. In the optimization of electrolyte concentration and flow rate as we have demonstrated here (Supplementary Fig. 6), a noticeable enhancement of FF in LSV curves is mainly attributed to the improved kinetics under the faster flow rate and higher concentration. We expect the FF to be further improved by developing redox couples with faster kinetics and engineering of the photoelectrode/electrolyte interface to facilitate charge extraction[14,48]. Second, because the SOEE of the SFB is very sensitive to the LSV behavior of the photoelectrode, the decreased FF of the GaAs photoanode would significantly alter its voltage matching with the redox couples in SFBs. On the other hand, the SOC swings will not create as much voltage mismatch in higher voltage photoelectrodes such as tandem III–V cell, because of the relatively small voltage shift through the *J–V* curves. Accordingly, a higher $V_{oc}$ photoelectrode is more likely able to achieve a better match and better SPUR of the SFB. (see Supplementary Fig. 11 for complete details).

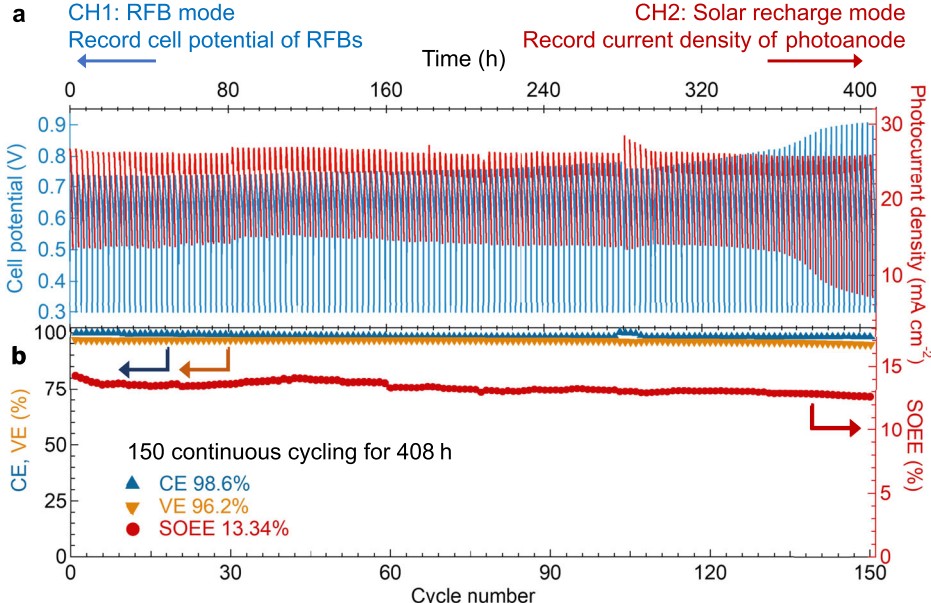

**Fig. 4 SFB charging/discharging cycling performance.** The SFB was integrated with SJ-GaAs photoanode and BTMAP-Vi/BTMAP-Fc redox couples. **a** Representative device cycling behavior showing the cell potential of SFB (blue curves), as well as the photocurrent density delivered by the GaAs photoanode (red curves). **b** Cycling efficiency plots of the integrated SFB device showing CE (blue triangles), VE (orange triangles), and SOEE (red circles). The SFB cycling was performed under one Sun solar illumination over 150 cycles with 0.2 M BTMAP-Vi/Fc redox couples in catholyte/anolyte and a flow rate of 60 mL min$^{-1}$. Each cycle started with 1.35 h of bias-free solar charging process followed by a galvanostatic discharging step at 11 mA until reaching the cutoff potential (0.3 V).

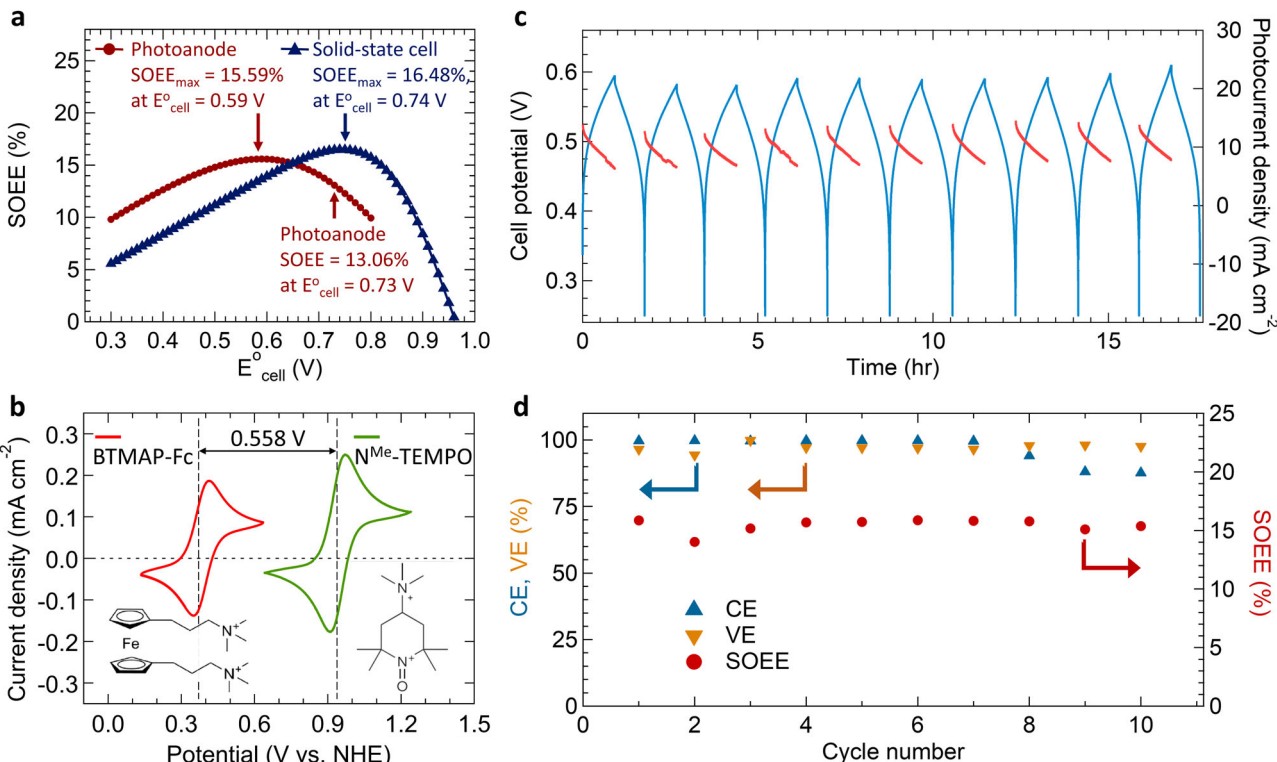

**Fig. 5 The potential matching simulation and the characteristics of SFB cycling.** An improved SFB with SJ-GaAs photoanode and BTMAP-Fc/N$^{Me}$-TEMPO redox couples. **a** The numerically calculated SOEE as a function of $E^0_{cell}$ by using the LSV data from the SJ-GaAs photoelectrode (red curve was simulated by the data from Fig. 3a) and solid-state SJ-GaAs cell (blue curve was simulated by the data from Fig. 1d). **b** The cyclic voltammograms of 5.0 mM BTMAP-Fc (red curve) and 5.0 mM N$^{Me}$-TEMPO (green curve) collected at a scanned rate of 10 mV s$^{-1}$ on a glassy carbon electrode in 1.0 M NaCl supporting electrolyte. **c** Cell potential (blue) and photocurrent density (red) vs. time of the integrated SFB device during cycling. **d** CE (blue triangles), VE (orange triangles) and SOEE (red circles) of the integrated SFB device over 10 cycles. The SFB cycling was performed with 0.1 M BTMAP-Fc/N$^{Me}$-TEMPO redox couples in catholyte/anolyte and a flow rate of 60 mL min$^{-1}$ over 10 cycles under one Sun solar illumination. Each cycle started with 36 min of bias-free solar charging process followed by a galvanostatic discharging step at 11 mA until reaching the cutoff potential (0.25 V).

Indeed, when the LSV curve of the SJ-GaAs photoanode is used for the SOEE-$E^0_{cell}$ simulation (red curve in Fig. 5a), the predicted optimized $E^0_{cell}$ is shifted to 0.59 V, which is significantly lower than the optimized $E^0_{cell}$ of 0.74 V predicted using the $J$–$V$ curve of the solid-state GaAs cell (blue curve in Fig. 5a). Therefore, the $E^0_{cell}$ of the BTMAP-Vi/Fc redox couples was actually too high for the actual $J$–$V$ performance of the GaAs photoanode. The simulated $SOEE_{ins}$-SOC curves for further comparison of the charging behavior by using the SJ-GaAs SFB with the $E^0_{cell}$ of 0.46, 0.56, and 0.66 V (Supplementary Fig. 12) revealed more uniformly higher $SOEE_{ins}$ values across various SOC levels by using the $E^0_{cell}$ of 0.56 V. In light of this, we further studied a RFB using the BTMAP-Fc and $N^{Me}$-TEMPO[50–52] redox couples in catholyte/anolyte that could deliver a better matched $E^0_{cell}$ of 0.558 V as demonstrated in Fig. 5b. An impressive average SOEE of 15.4% for 10 cycles of SFB cycling by using the BTMAP-Fc and $N^{Me}$-TEMPO redox couples (Fig. 5c), and the average CE and VE of 96.76% and 97.19% can be obtained, respectively (Fig. 5d). Unfortunately, the rather fast capacity decay of the RFB built with these redox couples (Supplementary Fig. 13) prevented us from demonstrating long-term cycling. Hopefully, this issue can be solved by investigating the capacity decay mechanism of the RFB, or by developing other suitable robust redox couples with similarly targeted potentials in future work.

## Discussion

Due to the rational $E^0_{cell}$ matching and operating condition optimization, this further improved SJ-GaAs SFB achieved an average SOEE of 15.4% and a SPUR of 67.6% (based on a PCE of 22.78% for the SJ-GaAs solar cell), which are the highest among all the SFBs with SJ photoelectrode reported so far[17,48,53,54]. This SOEE is even higher than that of the previously reported triple-junction III–V SFB despite the higher PCE of 26.1% for the III–V tandem cell[15] due to the lower SPUR (54.0%) achieved there than the current SJ-GaAs SFB. This is because the triple junction cell was not integrated with redox couples with the best matched $E^0_{cell}$ (predicted to be 1.72 V, as shown in Supplementary Fig. 14), which is limited practically by the thermodynamic water splitting potential in aqueous electrolyte (1.23 V). That SFB also did not operate in the optimal SOC range. These results and analyses show that comprehensive potential matching simulation provides the better procedure to design and charge the integrated SFB device, which is the most critical factor to improve both the SOEE and SPUR for a general integrated SFB system. Such improved analyses enabled us to achieve a better SFB performance using an SJ photoelectrode than what was previously achieved using a much more expensive triple junction photoelectrode[15]. The cost of SJ-GaAs cells can also be further reduced by utilizing epitaxial lift-off fabrication approach[30,32,41] or thin-film GaAs solar cells[55] in the future.

Furthermore, we summarize the SFB performance in representative previous reports in comparison with that presented herein in Fig. 6. Several key parameters are compared: SOEE (horizontal axis), the current density of the photoelectrode (vertical axis), demonstrated cycling lifetime (the radius of the circles). The solar cell structure of each work is marked by the symbols of the red triangle (for single junction) and green pentagon (for tandem junction), individually. The photoelectrodes, redox couples, and the corresponding energy capacity of SFB are displayed near each work. The pH of the electrolytes is also marked with the color of the data symbol. It can be clearly seen that the SJ-GaAs SFB device demonstrated in this work features the largest photocurrent density, outstanding continuous operation time, and one of the highest SOEEs (the highest SOEE among all the SJ photoelectrode based SFBs). There are several

key factors contributing to the efficiency and stability of this record-holding SFB device: First, the high efficiency of the back-illuminated SJ-GaAs photoelectrode with an unusual n–p–n sandwich design that is friendly for incorporation into liquid cells and neutral aqueous RFB electrolyte with robust BTMAP-Fc/Vi redox couples. Further, the effective protection of GaAs with ALD-TiO$_2$ coating in the more friendly neutral aqueous electrolyte. Most importantly, the rationally optimized potential matching and operation conditions of the SFB device.

In addition to the efficiency and stability, the solar charging photocurrent density is also a valuable metric for SFBs but has received much less attention so far. Because the operating current density of a typical RFB is usually quite high (>50 mA cm$^{-2}$), higher photocurrent density of the photoelectrode would result in higher redox couple concentrations (thus higher energy storage capacity) in SFBs and less geometrical area mismatch between the photoelectrode and RFB electrode in SFBs, which in turn could facilitate device engineering and lower the fabrication cost for practical SFB devices. In this regard, high solar charging photocurrent densities could be beneficial for practical SFBs, and therefore, a fruitful future direction could be investigating high-performance SJ solar cells[21,22] that can deliver high photocurrent densities but with relatively low photovoltages[56]. The current study also reveals the challenges that need to be addressed to achieve even higher SOEE and SPUR using this type of photoelectrodes: improving the fill factor under high photocurrent density conditions by optimizing the concentration and flow rate of the electrolytes, enhancing redox couple kinetics, and refining photoelectrode engineering[48], and the need for a repertoire of diverse and robust redox couples[19] with closely spaced formal potential values for more precise voltage matching[56].

In summary, we demonstrated a high performance and long lifetime integrated SFB system with a back-illuminated SJ GaAs photoelectrode and robust aqueous organic redox couples. The average SOEE of 15.4% using the BTMAP-Fc/$N^{Me}$-TEMPO redox couples sets a new record for the SFBs with SJ photoelectrodes. Compared with the previous SFB based on III–V triple-junction solar cells, the fabrication cost of the integrated SFBs in this work is significantly reduced and the lifetime is much longer, yet without sacrificing SOEE performance. More importantly, the rational potential matching simulation and comprehensive operating condition optimization enabled an excellent SOEE and SPUR that surpass the previous III–V triple-junction based SFB. The device operation optimization methods developed in this work can also serve as a general strategy for improving the performance of other integrated solar energy conversion and storage devices[3,4]. Compared with separate solar cell + battery devices, integrated SFBs made of cost-effective solar cells could have the benefits of lower cost due to the saving in the expensive maximum power point tracking and DC–DC conversion electronics[3,57], potentially higher efficiency, and convenient integrated thermal management in a compact device[14,15,17,56,58,59]. Our results not only demonstrate a new high-performance SFB but also shed new insights on how to design highly efficient SFBs based on more practical SJ solar cells that often have higher photocurrent densities but relatively lower photovoltages (than tandem solar cells). The success in further developing SFBs could enable practical off-grid electrification applications, such as solar home systems[56,57,60].

## Methods

**Fabrication of SJ-GaAs cell and photoelectrode**. The n-on-p configuration GaAs SJ cells were fabricated in a low-pressure metal–organic chemical vapor deposition (MOCVD) system (EMCORE D180, Agnitron Technology) at 615 °C. A 200-μm-thick n-GaAs substrate was diced into 10 mm × 10 mm square pieces, cleaned, followed by the deposition of a 200 nm of Ni/Au layer as the back metal contact.

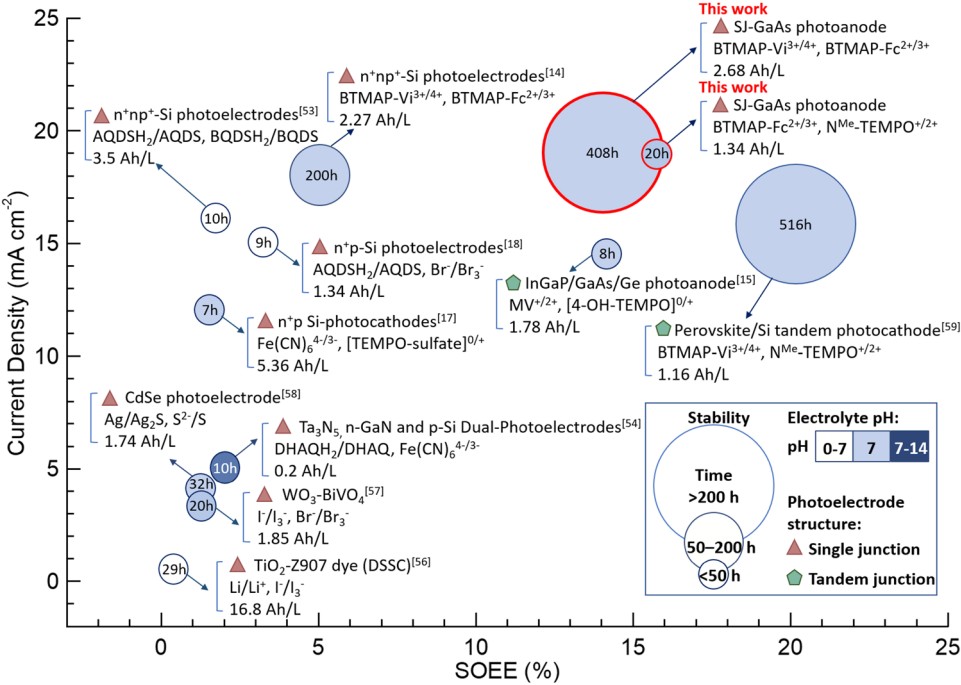

**Fig. 6 SFB performance in comparison with representative previous works**[14,15,17,18,53,54,60,65–67]**.** The number in the circle and the circle radius represent the demonstrated continuous cycling time (in an hour) and their corresponding range, respectively. The fill color of the circle shows the electrolyte pH range. The solar cell structure of each work is marked by the symbols of the red triangle (for single-junction) and green pentagon (for tandem junction), individually. The photoelectrodes, redox couples, and the corresponding energy capacity of SFB are displayed near each work.

Then, a 500 nm of n+-GaAs buffer layer (for lattice matching to the active junction)[61,62] and a 40 nm of p++-AlGaAs tunnel junction were grown on the substrate, subsequently. Next, a 100 nm of p+-GaInP as a back surface field, and then GaAs SJ n-on-p configuration followed by an (emitter)-on-p (base) active junction, 35-nm-thick n+-AlGaAs window and 500-nm-thick n+-GaAs contact layers were grown by low-pressure MOCVD. The group III precursor sources were trimethylgallium, trimethylaluminum, and trimethylindium, and the group V precursor sources were arsine and phosphine. Carbon tetrabromide and dimethyl telluride were used as precursors for p- and n-type dopants, respectively. The growth was carried out at a low pressure of 40 torr with a hydrogen flow rate of 28,000 sccm. Ni/Au (50/150 nm) finger grides was deposited on the n+-GaAs contact layer as the front electrode.

To fabricate the GaAs photoelectrode assembly, the GaAs solar cell was affixed onto a custom-made graphite plate, which has an open window of 9 mm × 9 mm at the center. Epoxy resin (Hysol 9460) was used to seal around the GaAs cell on the open window of the graphite, which can prevent electrolyte leakage and direct electrical contact between the GaAs cell and graphite plate. The light-harvesting surface and photoelectrode/electrolyte contact area were exposed without applying the epoxy. The active area of the SJ-GaAs photoelectrodes was calculated using calibrated digital images (as shown in Supplementary Fig. 3) in Photoshop. This configuration allows the GaAs photoanode to absorb light from one side (n+ window side) and form direct contact with liquid electrolyte on the other side (n+ substrate side). The Ni/Au electrode grids that act as carrier collector on the light-harvesting side were connected to a Cu foil using Ga/In eutectic alloy (Sigma-Aldrich) and then silver paste and sealed by epoxy resin.

**Solid-state and PEC characterization of SJ-GaAs cell.** Solid-state J–V performance of the GaAs cells was measured in a two-electrode configuration[63]. The LSV measurements were carried out using a Bio-Logic SP-200 potentiostat with a scan rate of 100 mV s−1 under AM 1.5 G one Sun (100 mW cm−2) illumination by a Newport Model 91191 Xenon arc lamp solar simulator. The illumination intensity of the solar simulator was calibrated by a Si photodiode (Thorlabs) before LSV measurements.

The PEC characteristics of the GaAs photoanode were measured using the integrated SFB device under solar cell mode in an N2 flush box by a Bio-Logic BP-300 potentiostat in a two-electrode configuration under one Sun illumination. The LSV measurements were performed with a scan rate of 100 mV s−1. The simulated solar illumination was provided by a Newport Model 67011 quartz tungsten halogen (QTH) solar simulator and guided by a branched flexible silica light guild (Taiopto Mems International Co., Ltd.) fed through an N2 flush box. The QTH solar simulator was calibrated by the same Si photodiode calibration cell to

generate the same value of current intensity as that measured under one Sun AM1.5 G illumination by the Newport 91191 simulator.

**EQE spectral measurements.** The EQE spectra were measured using a spectral response system (Enli Technology Co., Ltd. R3011). The GaAs solid-state solar cell was measured under monochromatic illumination, with wavelength ranging from 300 to 890 nm, at a spot area of 2 × 2 mm and a chopping frequency of 230 Hz, in a two-electrode configuration. The EQE signal was recorded at an applied potential of 0 V (short-circuit). The reflection spectrum was measured by a UV–vis–NIR spectrometer (JASCO ARN-733) and scanned from the wavelength of 300–890 nm with an integrating sphere at a noise level of 0.002%.

**Electrochemical measurements of the redox couples.** CV measurements were performed using a Bio-Logic SP-200 potentiostat. A Pt coil electrode (0.5 mm diameter, BASi) and a saturated calomel electrode (SCE, CH Instruments) was used as the counter and a reference electrode, respectively. The working electrode was a 3 mm diameter glassy carbon disk electrode (MF-2012, BASi), which was polished using 0.3 and 0.05 μm alumina slurry and washed by deionized water (Milli-Q, 18.2 MΩ cm) and methanol. An electrochemical cleaning procedure with 1 M Na2SO4 aqueous solution (with 1 mM potassium ferrocyanide as internal reference) was used to further clean the surface of the glassy carbon electrode, which was performed by sweeping the potential of glassy carbon working electrode between −1.0 and 1.5 V vs. SCE at 100 mV s−1 until the peak separation of ferrocyanide/ferricyanide redox couple reaches ca. 60 mV. Then 5 mM of bis((3-trimethylammonio)propyl)ferrocene dichloride (BTMAP-Fc), 5 mM of bis(3-trimethylammonio)propyl viologen tetrachloride (BTMAP-Vi), and 5 mM of 4-trimethylammoinium-TEMPO (NMe-TEMPO) were both with 1.0 M NaCl, were used as the supporting electrolytes, which were purged with argon for 10 min before the CV measurements. CV was scanned at various scan rates of 10, 100, 200, 400, 600, 1000, and 2000 mV s−1.

**Fabrication of RFB and SFB device.** A custom-made zero-gap device was used for both RFB and SFB measurements[15]. Graphite plates (1/8-in. thickness, MWI) were used as the current collector for RFB devices and the cathode side of SFB devices. The modified graphite plates in the GaAs photoanode assemblies were used as the current collector at the anode side of SFB devices. Graphite felt electrodes (GFD 3 EA, SIGRACELL®) (20 × 20 mm) was heated at 400 °C in the air for 24 h before being used as inert electrodes on both sides of the cell. A 25 × 25 mm Selemion DSV membrane (Ashahi Glass Co., Ltd.,) was pretreated by soaking in 1.0 M NaCl for 24 h before being used as the anion-exchange membrane in the cell. Four pieces of custom-made PTFE sheets (0.04-in. thickness) were used as gaskets. These

components were bolted together with eight #10–24 bolts. A peristaltic pump (Cole-Parmer Masterflex L/S) was used to circulate the electrolytes between the electrolyte reservoirs (contained in 15 mL polypropylene centrifuge tubes) and the SFB/RFB cells via PharMed BPT tubing.

**RFB device characterizations**. All RFB and SFB measurements were carried out in a custom modified $N_2$ flush box (Terra Universal) with continuous $N_2$ purging. 5.0 mL solution of 0.2 M BTMAP-Fc and 5.0 mL solution of 0.2 M BTMAP-Vi, both with 1.0 M NaCl as the supporting salt, were used as anolyte and catholyte, respectively. Both BTMAP-Fc and BTMAP-Vi were purchased from the Tokyo Chemical Industry Co., Ltd. and used directly. The $N^{Me}$-TEMPO was synthesized following a previous report[50]. The electrolyte flow rate was set from 20 to 120 mL min$^{-1}$ for RFB measurements. Galvanostatic cycling tests were carried out using a Bio-Logic BP-300 potentiostat at desired constant current densities with 0.3 and 1.1 V as the bottom and top potential limits, respectively. A 10 s rest period at open-circuit voltage was employed between each half cycle. The potentiostatic capacity of the RFB was determined by galvanostatic charging/discharging followed by a potential hold at cut-off potentials until the current density reached 1 mA cm$^{-2}$.

**SFB device characterizations**. Totally, 5.0 mL solution of BTMAP-Vi/Fc with concentrations of 0.1 and 0.2 M in 1.0 M NaCl, or 5.0 mL solution of BTMAP-Fc/ $N^{Me}$-TEMPO with concentrations of 0.1 M in 1.0 M NaCl were used as the catholyte/anolyte. The electrolyte flow rate was controlled at 40, 60, and 80 mL min$^{-1}$ for the SFB cycling tests. A dual-channel Bio-Logic BP-300 potentiostat was used for the SFB cycling tests. To characterize the charging–discharging behaviors of the integrated SFB device, one potentiostat channel (CH1) was configured as the RFB mode to monitor the potential between the two carbon felt electrodes; the other potentiostat channel (CH2) was configured as solar recharge mode to monitor the charging photocurrent (Fig. 1c).

During the solar charging process, the GaAs photoelectrode was illuminated by one Sun simulation (as described in the PEC characterization section) without applying external bias. A 1.35 h time limit was used to control the SOC below ca. 54%. During the discharging process, the illumination was blocked by a beam shutter, and a discharging current intensity of 11 mA was applied by CH1 until the cell potential reached 0.3 V. The dual-channel potentiostat and the beam shutter of the solar simulator were synchronized and controlled by CH1 and a custom-made electronic control box to enable automated long-term SFB cycling measurements.

**Potential match calculation and simulations**. To optimize the potential match between the photovoltage of GaAs photoelectrode and the formal cell potential of SFB ($E_{cell}^0$), we carried out SOEE simulation with different hypothetical $E_{cell}^0$. Note that $E_{cell}^0$ generally remains constant with given anolyte/catholyte combination, but the actual cell potential of SFB ($E_{cell}$) changes with SOC and needs to be calculated by the Nernst equation:

$$E_{cell} = E_{cell}^0 - \frac{RT}{nF}\ln\left[\frac{1-SOC}{SOC}\right]^2 + IR_{DC},\qquad (5)$$

where $R$ is the universal gas constant, $T$ is temperature, n is the number of electrons transferred in a redox reaction, $F$ is Faraday constant, $I$ is the applied current, and $R_{DC}$ is the DC series resistance of SFB under RFB mode[64].

The solar conversion efficiency of SFBs at specific SOCs was quantitatively assessed by SOEE$_{ins}$ (Eq. 4). The equations for calculating the CE, VE, and EE are

$$CE = \frac{Q_{discharge}}{Q_{charge}} = \frac{\int I_{out}dt}{\int I_{operating}dt},\qquad (6)$$

$$VE = \frac{\bar{V}_{discharge}}{\bar{V}_{charge}} = \frac{\frac{\int V_{out}dt}{\int dt}}{\frac{\int V_{operating}dt}{\int dt}},\qquad (7)$$

$$EE = \frac{E_{discharge}}{E_{charge}},\qquad (8)$$

where $I_{out}$ is the discharging current and $V_{out}$ is the cell potential extracted from data during the discharging process. Therefore, the qualitative relationship between SOEE$_{ins}$ and $E_{cell}$ (or SOC) at specific $E_{cell}^0$ can be obtained (Fig. 3b, Supplementary Fig. 11, and Supplementary Fig. 12). The overall SOEE of SFB charged in a specific SOC range ($x$ to $y$, usually from 1% to 99% if not specified otherwise) was then calculated as the integral average of SOEE$_{ins}$ with respect to SOC.

$$SOEE = \frac{\int_x^y SOEE_{ins}(SOC)dSOC}{\int_x^y dSOC}.\qquad (9)$$

By repeating the calculation described above with different $E_{cell}^0$ and a 10 mV interval, the qualitative relationship between SOEE and $E_{cell}^0$ can be obtained (Figs. 1d and 5a and Supplementary Fig. 14).

## Data availability

All data needed to evaluate the conclusions in the paper are present in the paper and/or the Supplementary Information. Additional data related to this paper may be requested from the corresponding authors upon reasonable request. Source data are provided with this paper.

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

## Acknowledgements

This research is supported by the King Abdullah University of Science and Technology (KAUST) Office of Sponsored Research under award No. OSR-2017-CRG6-3453.02. C. H.L. and J.H.H. are supported by the KAUST baseline fund for the design and fabrication of the single-junction GaAs cells. The authors thank Mr. Hongyuan Sheng for performing the XPS analysis on the GaAs photoelectrodes.

## Author contributions

H.C.F., W.L., Y.Y., and S.J. designed the experiments. H.C.F. and W.L. fabricated the SFB devices, and H.C.F. and Y.Y. carried out the electrochemical measurements. A.V. assisted with the evaluation of the redox couples. H.C.F., C.H.L., and J.H.H. fabricated the single-junction GaAs cells. H.C.F., W.L., J.H.H., and S.J. wrote the paper, and all authors commented on the paper.

## Competing interests

The authors declare no competing interests.
