## [Peer Review File · Nature Communications]

REVIEWER COMMENTS

Reviewer #1 (Remarks to the Author):

This manuscript presented a single junction GaAs photoelectrode enable a solar-flow battery. The proposed consideration towards efficiency simulation is interesting which should be of interest to the readers in relevant fields. I have some detailed comments for the authors to consider and to further improve the paper quality.

1. The solar flow battery was designed to simultaneously harvest and store solar energy for standalone electrification in remote areas as stated in the manuscript. However, for practical application without requiring mobility, is it necessary to monolithically integrate photovoltaics with redox flow batteries? The design of monolithic solar flow batteries is complicated and their fabrication is costly. Moreover, it would be challenging to enhance the long-term stability of monolithic devices for practical application as the photovoltaic unit has to be in direct contact with corrosive electrolyte.

2. Can the authors comment on the competitiveness of solar flow battery in terms of overall cost in comparison with the other technologies on the market such as photovoltaic combined with lithium battery or electrolysis for hydrogen generation?

3. What is the active area of the solar flow battery in this work? And can the authors comment on the scalability?

4. The TiO₂ protection layer on GaAs photoelectrodes and the neutral pH electrolyte were used in previous literature on GaAs cells to enable the stability. If authors wish to highlight it, I suggest to explain more on this part.

5. When the author simulated the matching between solar cells and flow batteries with the proposed equation, the resistance part plays important role in it, especially considering the TiO₂ layer was applied between the two parts with high resistance. This generates another question towards the core part of this single-junction GaAs SFB. The author also highlighted that with single solar cell, the current density would be higher than that of the previous reports. While the larger influence on the voltage decay and more importantly, energy loss, would drastically reduce overall performance of this system. I would suggest the authors to consider and discuss this issue carefully.

6. The Authors claimed the high areal capacity of the flow batteries, while the energy density of the energy storage system is very important in the applications. It may be better to directly show the energy density (mAh L⁻¹) of the redox-flow batteries in this SFB (Figure 2)?

7. In addition to the attempts to optimize the voltage and SOC of the RFB, more stable platform of the energy storage devices at certain potential would be more helpful to improve the matching and overall efficiency of the system. Authors may also comment on this direction.

Reviewer #2 (Remarks to the Author):

In this manuscript, Fu et al., have demonstrated a solar flow battery (SFB) utilizing a single junction GaAs solar cell. In their demonstration, they report solar-to-electricity conversion efficiency (SOEE) of 13.3%, with a continuous cycling performance of 408 h without much degradation in the performance. Although the results are interesting, it is hard to understand the practical significance of the proposed device. For instance, the authors have used n-type GaAs substrates to fabricate solar cells and claim it to be cheaper compared to p-type GaAs, however the GaAs substrates are much costlier compared to silicon which is widely used in solar cells.

In summary, the results are original and could be considered for a publication after the following issues are addressed.

1. The authors have highlighted the usage of single junction solar cell over a multijunction solar cell for coupling with flow battery. However, in practicality any single junction solar cell could be coupled with the flow battery comprising redox couples with redox potential close to voltage output of the solar cell? The authors should explain the rationale of choosing a single junction solar cell and coupling with redox couples with lower redox potential which would limit the energy density of the flow battery.
2. In page no. 8, should $EFc0$ be 0.328 V or 0.382 V in the equation 2?
3. In page no. 9, the authors mentioned that, higher flow rate of electrolyte causes destruction of the photo electrode. Is this due to the adhesion problem of the protective materials? Please explain the reason for the destruction of the photoelectrode materials at higher flow rate.
4. The authors should discuss the energy density of the SFB and make a comparison with the literature?
5. In Fig 3 (a), the result suggests that at higher State of Charge (SOCs) of the SFB there is a drastic reduction in the current. Is this reduction arising from the solar cells or due to the redox couples used? In a recent report (Nature Materials. 2020 Jul 13:1-6.), the authors demonstrated more than 80% of the current is retained even at 90% SOC.
6. In page no 17, was n-GaAs substrate diced into 100 mm × 100 mm square pieces or 10 mm x 10 mm square pieces?

Reviewer #3 (Remarks to the Author):

Solar flow battery has become a rapid trend for the solar cell and flow battery communities. Given the poor chemical resistance of solar cells, the flow chemistry in solar flow battery is oftentimes non-corrosive and mild neutral. Because of this limit, the concentration of active materials is always very low, which is actually at least three orders of magnitude lower than that of commercial vanadium flow battery. This has caused lots of energy waste for the pumping system. For instance, given a flow rate of 50 ml/min, the energy stored by SFB is 1/1000 of that by traditional FB that is charged by traditional solar cell. In this sense, the technological significance of SFM is rather poor. Having that said, it is still of great value to explore the novel integrated photo-electrochemistry at the interface. Finding new chemistry will be always valued for fundamental acquisition of new knowledge. In this work, the authors adopted existing systems of redox couples and GaAs photo absorber. It is important to be aware of the toxicity of As which is not in fact popular in solar cell field. This work presents the innovative engineering by integrating two parts from existing knowledge and technical research database - it is for sure remarkable optimisation experiments were conducted to achieve the reported high performance. Yet scientifically, very limited new knowledge can be obtained from this work.

Reviewer #1 (Remarks to the Author):

This manuscript presented a single junction GaAs photoelectrode enable a solar-flow battery. The proposed consideration towards efficiency simulation is interesting which should be of interest to the readers in relevant fields. I have some detailed comments for the authors to consider and to further improve the paper quality.

We would like to thank Reviewer #1 for his/her positive comments and recommendation for publishing this work in *Nature Communications*. We have addressed the comments by the reviewer and revised the manuscript accordingly.

1. The solar flow battery was designed to simultaneously harvest and store solar energy for standalone electrification in remote areas as stated in the manuscript. However, for practical application without requiring mobility, is it necessary to monolithically integrate photovoltaics with redox flow batteries? The design of monolithic solar flow batteries is complicated and their fabrication is costly. Moreover, it would be challenging to enhance the long-term stability of monolithic devices for practical application as the photovoltaic unit has to be in direct contact with corrosive electrolyte.

Response: We agree with the reviewer that, just for the sake of storing the solar energy, separate PV cells + battery could also work, it is “not necessary” to monolithically integrate PV cells with redox flow batteries. However, there are several benefits for using integrated SFB over separated PV + battery:

(I) Integrated SFBs have lower overall system cost. When the integrated SFBs are designed well with rational potential matching between the PV and RFB components, the rather expensive maximum power point tracking (MPPT) and DC-DC conversion electronics are no longer needed in the charge controller, which can significantly bring down the cost of the overall system. In addition, the cost for electrical and structural balance of system (BOS) of integrated SFB will also be reduced, since a SFB does not need additional electrical cables to connect the PV module and RFB module or additional enclosure and mechanical support materials that are necessary for separate PV + RFB devices. The exact cost reduction percentage depends on the relative cost of the PV cells, RFB modules, charge controllers, BOS, and other components, and would be difficult to calculate exactly for the current case since GaAs PV and aqueous organic RFB are not yet commercially deployed and reliable cost information is not available. However, it is clear that, overall, integrated SFB devices will be less costly than separate PV + RFB devices with similar parameters.

(II) Integrated thermal management. In the SFB device, the flowing aqueous electrolyte is in direct contact with the photoelectrode so that excess heat can be readily removed from photoelectrode to ensure a stable solar performance. The flowing electrolytes can also significantly increase the safety level of the SFB device due to the fire-retardant nature of aqueous electrolyte.

(III) Well designed SFBs can have better system efficiency. By assuming the same RFB module efficiency, the solar power conversion utilization ratio (*SPUR*, defined as the ratio between the *SOEE* of the SFB and the *PCE* of the photoelectrode) of separated PV+RFB system is limited by the efficiency of MPPT and DC-DC conversion electronics, which is usually between 82%-96%. Most importantly, with the rational potential matching and operating condition optimization in this work, the SJ-GaAs SFB has achieved a *SPUR* of 67.6%, which is much better than the previously reported SFB with III-V photoelectrode (54.0%). The modeling approach and the design principles presented in this work would allow us to make more optimally designed and matched SFBs that could maximize the *SPUR* to reach better overall system efficiency.

As the Reviewer recognized, indeed we need to carefully engineer the photoelectrodes (and the electrolytes) so that they are stable in the electrolyte solution. As we showed here, using neutral pH electrolyte is quite important and properly protecting the semiconductors is the key. We have readily achieved operational stability of more than 500 hours using TiO₂-protected GaAs electrode, which is significantly improved from the 10-20 hour lifetime commonly demonstrated for photoelectrochemical water splitting devices made with III-V photototelectrodes.

Of course, significant developments are always needed to improve any nascent technologies to make them competitive with established ones to realize their potential benefits (as discussed above). That is why we are working on this: it is very important to design and fabricate efficient photovoltaic (PV) cells and photoelectrodes with suitable protection/modifications while developing the suitable robust redox couples with optimal potentials, that address the issues of cost, cycling stability, voltage match, and slow carrier transfer kinetics across the photoelectrode/electrolytes interface, so we further enhance the performance and stability of SFBs.

We have added a brief discussion of these benefits of SFBs in the conclusion paragraph in page 18 and cited some additional references to support these points.

2. Can the authors comment on the competitiveness of solar flow battery in terms of overall cost in comparison with the other technologies on the market such as photovoltaic combined with lithium battery or electrolysis for hydrogen generation?

Response: We think there are two layers to this question, one is the comparison between integrated SFB with separate PV+battery, the other is the comparison between RFB, lithium ion battery and then with PEC hydrogen generation.

The responses to the question above essentially answered the question about integrated device vs. separate device. We further note that the costs of both PV module and RFB modules are expected to decrease due to the continuing development and increasing deployment of both PV and RFB technology. These developments could further favor integrated SFB over separated PV+RFB that has higher cost of charge controller, electrical and structural BOS taking up larger fraction of the total system cost.

In terms of the comparison between RFB and lithium ion battery, Li-ion batteries have higher energy densities, which is favorable for portable electronics, but at the larger scale, they are not cost competitive with RFBs. The strength of the RFBs is the decoupled energy and power scale and thus the lower cost at larger scale. For storing large amount of solar energy, energy density does not matter too much, unlike in the case of portable electronics or even electric vehicles for which the weight or volume are at premium, instead, the total capacity and average cost per capacity are the more crucial factors. Therefore, at larger scale, RFBs will win over Li-ion batteries, and integrated SFBs will win over separate PV + Li-ion batteries.

The attraction of PEC electrolysis is the production of clean hydrogen fuel, but as we discussed in the introduction, if such solar hydrogen will be used to generate the electricity back using fuel cells, the SFBs here essentially fulfil the same function (round trip solar energy harvest, storage and delivery of electricity), but will be more efficient because we avoid the sluggish kinetic steps of electrocatalysis (HER, OER, and then ORR, HOR) on electrode surfaces. In fact, let's use some published examples to compare: after years of developments and optimization at JCAP (the Joint Center for Artificial Photosynthesis), the cited Ref. 42 (Monolithic Photoelectrochemical Device for Direct Water Splitting with 19% Efficiency, *ACS Energy Lett.* **2018**, 3, 1795–1800) show 19.3 % solar-to-hydrogen (STH) efficiency (for less than 1 hour stability) using a **double tandem junction III-V** solar cell with a PCE of about 32% (or 18.5% STH for 20 hour stability). In comparison, we have already achieved 15.4 % SOEE using a **single junction GaAs solar cell** with a PCE of 22.78%. Furthermore, when the hydrogen is used to regenerate electricity in the separate fuel cells that demand significant additional cost, more efficiency will still be lost. Even though it is difficult to compare head-to-head since we do not have the same underlying PV cells to work with, we already can do shows that SFBs are more efficient and affordable than PEC hydrogen generation, especially if what one needs is the electricity in the end.

3. What is the active area of the solar flow battery in this work? And can the authors comment on the scalability?

Response: We thank the reviewer for the comment. The active area of each SJ-GaAs photoelectrode was calculated using calibrated digital images (as shown in Supplementary Fig. 3) in Photoshop. The active area of the photoelectrodes used was controlled to c.a. 0.5 cm². More detailed description has been added in P.19 of the manuscript in the Method section.

P.19 of the manuscript of the Method section:

The light harvesting surface and photoelectrode/electrolyte contact area were exposed without applying epoxy. The active area of the SJ-GaAs photoelectrodes were calculated using calibrated digital images (as shown in Supplementary Fig. 3) in Photoshop.

The scale of the SFB device obviously depends on the scale of the PV solar cells used, and the currently demonstrated SFB devices use typical prototypic solar cell devices with the device area that is in line with what is commonly reported in PV cell research literature (0.1 to 1 cm²).

4. The TiO₂ protection layer on GaAs photoelectrodes and the neutral pH electrolyte were used in previous literature on GaAs cells to enable the stability. If authors wish to highlight it, I suggest to explain more on this part.

Response: We thank the reviewer for the suggestion. In the revised version of the manuscript, we have further highlighted the function of the Ti/TiO₂/Pt protection layer for the SFB and added more discussion on the post-mortem X-ray photoelectron spectroscopy (XPS) analysis originally presented in the Supplementary Information to elucidate the failure mechanism of the surface protection layer of Ti/TiO₂/Pt in several places as following:

P.6 of the manuscript:

Note that, a protection layer of Ti/TiO₂/Pt was deposited on the electrolyte contacting surface of the photoelectrode as illustrated in Figure 1a. The *J-V* performance of the GaAs cells with and without Ti/TiO₂/Pt layer is shown in the Supplementary Fig. 1. The DC series resistance (R_{DC}) of the GaAs cell was calculated according to the J-V curve, which only showed a less than 0.3% increase after depositing the metal oxide layer.

P.7 of the manuscript:

Previous reports have shown that the introduction of a TiO₂ surface protection layer on III-V semiconductors can effectively protect the photoelectrodes from photocorrosion. Therefore, we further deposited a TiO₂ thin film by ALD on the electrolyte contacting surface (see Figure 1a) of GaAs cells to serve as the protection layer and enable stable long-term operation.

P.10 of the manuscript:

Post-mortem optical imaging (Supplementary Fig. 3c to 3e) and X-ray photoelectron spectroscopy (XPS) surface analysis (Supplementary Fig. 8) were further performed to understand the failure mechanism of the surface protection layer of Ti/TiO₂/Pt. They suggested that the TiO₂ protection layer (together with the Pt layer on top) disappeared after cycling for the case of the 80 mL min⁻¹ flow rate, but remained mostly intact for the cases of lower flow rates. The disappeared Pt XPS signal and the newly emerged Ga and As signals after cycling at the 80 mL min⁻¹ flow rate indicated that the Ti/TiO₂/Pt protection layer was peeled off during the SFB cycling test to expose the vulnerable GaAs substrate to electrolyte. Therefore, the observed SFB device performance decay was likely caused by such mechanical damage under the high flow rate. Therefore, in the remaining SFB measurements of this work, the electrolyte flow rate was set to 60 mL min⁻¹ to balance the efficiency and stability concerns.

P.14 of the manuscript:

The XPS analysis of the SJ-GaAs photoanode surface after 150 charge/discharge cycles (Supplementary Fig. 10) revealed diminished Pt and Ti signals and newly emerged Ga and As signals, which suggested that the Ti/TiO₂/Pt protection layer was significantly damaged or

peeled off during the long operation period to expose the vulnerable GaAs substrate to electrolyte.

5. When the author simulated the matching between solar cells and flow batteries with the proposed equation, the resistance part plays important role in it, especially considering the TiO₂ layer was applied between the two parts with high resistance. This generates another question towards the core part of this single-junction GaAs SFB.

Response: Thank you for this question. First, the simulation about the voltage match uses the experimental J-V curves measured from the actual solar cell device that incorporated the TiO₂ layer already, therefore, the resistance (if any) was already included in the consideration.

Second, we agree with the reviewer that there could be some DC series resistance (R_{DC}) introduced after depositing the TiO₂ layer. However, there have been many previous reports on using TiO₂ or other metal oxide layers to protect photoelectrodes (as discussed and cited in the responses to the question above) with thickness commonly up to a few hundred nanometers, and the resistivity as a function of the thickness behaviors of the ALD-TiO₂ protection layer as well as the stability have been widely studied (for example, the cited Ref. 38, DOI: 10.1126/science.1251428). In this work, a 80-nm-thick TiO₂ layer was deposited by ALD following a thin layer of Ti (5 nm) to promote adhesion and another thin layer of Pt (10 nm) was deposited on TiO₂ to enhance charge extraction at the photoelectrode/electrolyte interface. Furthermore, we added the measured J-V performance of the GaAs solar cells with and without the Ti/TiO₂/Pt layer as shown in the new Supplementary Fig. 1 (also reproduced below). The DC series resistance (R_{DC}) was calculated according to the J-V curves, which showed less than 0.3% increase after depositing these layers.

New Supplementary Figure 1. J-V performance of the solid-state SJ-GaAs cells with and without the Ti/TiO₂/Pt (5/80/10 nm) layer on the solar cell surface.

5. The author also highlighted that with single solar cell, the current density would be higher than that of the previous reports. While the larger influence on the voltage decay and more importantly, energy loss, would drastically reduce overall performance of this system. I would suggest the authors to consider and discuss this issue carefully.

Response: We thank the reviewer for this comment. We agree that the SJ-GaAs SFB had a drastic reduction in current with the BTMAP-Vi/Fc redox couples, which reduced the overall performance of the SFB. The reason for the lower operation current ($I_{operating}$) at higher SOC is illustrated in Figure 3a (reproduced below for convenience): by overlaying the I-V curves of the RFB at the different SOC (1 to 99% SOC) and the I-V curve of SJ-GaAs photoanode at 50% SOC, the operation points of the SFB can be found at the intersection points. The $I_{operating}$ was very sensitive to the LSV behavior of the photoelectrode at different SOCs.

Fig. 3. a) Estimation of $SOEE_{ins}$ for SFB built with SJ-GaAs photoanode and BTMAP redox couples.

In fact, according to the simulation, the E_{cell}^0 of the BTMAP-Vi/Fc redox couples was actually too high for the actual GaAs photoanodes, even though it was optimal based on the solid-state GaAs solar cells. In the revised manuscript, we studied a new SJ-GaAs SFB by using the better matched of N^{Me} -TEMPO and BTMAP-Fc redox couples that could deliver the E_{cell}^0 of 0.558 V and demonstrated further improved SOEE for this SFB. These results are shown in the new Figure 5.

But this bring a broader and general discussion about the comparison between single junction solar cells with lower V_{oc} and tandem solar cells with higher V_{oc} : Because the $SOEE$ of the SFB is very sensitive to the LSV behavior of the photoelectrode, the decreased FF of the GaAs photoanode would significantly alter its voltage matching with the redox couples in $SFBs$ (see the red curve in Supplementary Fig. 11a, reproduced below). In contrast, the same SOC swing will not create as much voltage mismatch in higher voltage cells such as tandem III-V cell (see the orange curve in Supplementary Fig. 11a), because of the smaller *relative* voltage shift through the $J-V$ curve. Accordingly, the instantaneous SOEE is much less sensitive to the

SOC for a higher V_{oc} cell (Supplementary Fig. 11b), therefore tandem solar cells with higher V_{oc} are more likely to achieve better voltage match and higher $SPUR$ of the SFB. This discussion is now added to page 15 of the revised manuscript and this new Supplementary Fig. 11 and the associated discussion was added in Supplementary Information.

New Supplementary Figure 11. Overlaid hypothetical current-potential behaviors between RFBs and two different photoelectrodes with different V_{oc} . **a** J - V performance of the SJ-GaAs (red) and triple junction III-V (orange, replotted from the data reported in ref.15) photoanode measured at solar cell mode overlaid with the I - V curves of RFBs at different SOC simulated with the respective optimally matched E_{cell}^0 of 0.59 V (green lines) and 1.72 V (blue lines). The SJ-GaAs photoelectrode were measured at the flow rate of 60 mL min⁻¹ under one Sun illumination at 50% SOC. **b** Normalized $SOEE_{ins}$ as a function of SOC for both solar cells.

The voltage at the maximum power point of the photoelectrode (V_{MPP}) should match as closely as possible with the swing range of $E_{cell}(SOC)$ to achieve a high $SOEE$. In Supplementary Fig. 11a, the V_{MPP} of the SJ-GaAs and triple junction III-V photoelectrodes of 0.61 V and 1.73 V, respectively, are marked as triangles. By overlaying the I - V curves of the RFB at the different SOC (1 to 99%), the swing range of $E_{cell}(SOC)$ by the optimally matched E_{cell}^0 of 0.59 V (for SJ-

GaAs) and 1.72 V (for triple junction III-V) can be mapped. In the lower voltage SJ-GaAs cell, the SOC swing creates large mismatch at extreme SOC levels (the red curve in Supplementary Fig. 11a). In contrast, the same SOC swing will not create as much mismatch in higher voltage tandem III-V cell (the orange curve in Supplementary Fig. 11a), because of the smaller *relative* voltage shift through the *J-V* curve. Accordingly, Supplementary Fig. 11b shows the instantaneous SOEE ($SOEE_{ins}$) is much less sensitive to the SOC for a higher V_{oc} cell, therefore tandem solar cells with higher V_{oc} are more likely to achieve better voltage match and higher *SPUR* of the SFB.

6. The Authors claimed the high areal capacity of the flow batteries, while the energy density of the energy storage system is very important in the applications. It may be better to directly show the energy density (mAh L⁻¹) of the redox-flow batteries in this SFB (Figure 2)?

Response: We agree that the energy density is an important factor for SFB. We had in fact already shown the capacity (mAh L⁻¹) of the SFB (before and after 150 cycles) in Supplementary Fig. 5 and Supplementary Fig. 9c. From this information and the voltage of the SFB, the energy capacity can be estimated to be 1.30 Ah L⁻¹ (the SFB was charged at an optimized SOC range from 0 to 54% in this work). The SFB showed a galvanostatic-potentiostatic charge/discharge capacity of 2.41 Ah L⁻¹ and an energy density of 2.05 Wh L⁻¹, which was mentioned in page 9 and 12 of the manuscript.

To further emphasize this point, we have now added the parameter of the capacity in the Figure 6 (the previous Figure 5) which summarizes the SFB performance in representative previous reports by comparing several key parameters: *SOEE* (horizontal axis), current density of the photoelectrode (vertical axis), demonstrated cycling lifetime (the radius of the circles), the pH of the electrolytes is also marked with the color of the data symbol and energy density value is noted for each entry.

Updated Fig. 6. SFB performance of representative previous demonstrations in comparison with this work. The number in the circle and the circle radius represent the demonstrated continuous cycling time and their corresponding range, respectively. The fill color of the circle shows the electrolyte pH range. The solar cell structure of each work is also represented by the symbols of triangle (for single junction) and pentagon (for tandem junction). The redox couples and the corresponding capacity of RFB are displayed near each work.

7. In addition to the attempts to optimize the voltage and SOC of the RFB, more stable platform of the energy storage devices at certain potential would be more helpful to improve the matching and overall efficiency of the system. Authors may also comment on this direction.

Response: Thanks for the suggestion. Indeed as discussed, when we use the actual $J-V$ curve of the GaAs photoanode, the simulation predicts an optimized E_{cell}^0 of 0.59 V, which is significantly lower than the E_{cell}^0 of 0.74 V predicted using the $J-V$ curve of the solid-state GaAs cell. Since the original submission, we have further studied another SJ-GaAs SFB by using the N^{Me}-TEMPO and BTMAP-Fc redox couples that could deliver a better matched E_{cell}^0 of 0.558 V and achieved a higher SOEE of 15.4% cycled for at least 10 cycles for this improved SFB. This higher SOEE is now even higher than the previous 14.1% achieved using triple junction III-V solar cells. These new results are added as the new Figure 5 and the new discussion in page 14-16.

New Fig. 5. The simulated SOEE - E^0_{cell} curve, CV, and cycling performance of a SFB device with SJ-GaAs photoanode and BTMAP-Fc/ N^{Me} -TEMPO redox couples. **a** The numerically calculated SOEE as a function of E^0_{cell} (red curve) by using the LSV data from the SJ-GaAs photoelectrode (the data from Figure. 3a) and solid-state SJ-GaAs cell (blue curve, the same as Fig. 1d). **b** The cyclic voltammograms of 5.0 mM BTMAP-Fc and 5.0 mM N^{Me} -TEMPO collected at a scanned rate of 10 mV s^{-1} on a glassy carbon electrode in 1.0 M NaCl supporting electrolyte. **c** Cell potential and photocurrent density of the intergrade SFB device during cycling. **d** CE (blue triangles), VE (yellow triangles) and SOEE (red circles) of the integrated SFB device over 10 cycles. The SFB cycling was performed with 0.1 M BTMAP-Fc/ N^{Me} -TEMPO redox couples in catholyte/anolyte and a flow rate of 60 mL min^{-1} over 10 cycles under one Sun solar illumination. Each cycle started with 36 min of bias-free solar charging process followed by a galvanostatic discharging step at -11 mA until reaching the cutoff potential (0.25 V).

Indeed, when the LSV curve of the SJ-GaAs photoanode is used for the SOEE- E^0_{cell} simulation (red curve in Figure 5a), the predicted optimized E^0_{cell} is shifted to 0.59 V, which is significantly lower than the optimized E^0_{cell} of 0.74 V predicted using the J - V curve of the solid-state GaAs cell (blue curve in Figure 5a). Therefore, the E^0_{cell} of the BTMAP-Vi/Fc redox couples was actually too high for the actual J - V performance of the GaAs photoanode. The simulated $SOEE_{ins}$ -SOC curves for further comparison of the charging behavior by using the SJ-GaAs SFB with the E^0_{cell} of 0.46, 0.56 and 0.66 V (Supplementary Fig. 12) revealed more

uniformly higher $SOEE_{ins}$ values across various SOC levels by using the E^0_{cell} of 0.56 V. In light of this, we further studied a RFB using the BTMAP-Fc and N^{Me} -TEMPO redox couples that could deliver a better matched E^0_{Cell} of 0.558 V as demonstrated in Figure 5b. An impressive average $SOEE$ of 15.4% for 10 cycles of SFB cycling by using the BTMAP-Fc and N^{Me} -TEMPO redox couples (Figure 5c), and the average CE and VE of 96.76% and 97.19% can be obtained, respectively (Figure 5d). Unfortunately, the rather fast capacity decay of the RFB built with these redox couples (Supplementary Fig. 13) prevented us from demonstrating long-term cycling. Hopefully, this issue can be solved by investigating the capacity decay mechanism of the RFB, or by developing other suitable robust redox couples with such targeted potentials in future work.

Reviewer #2 (Remarks to the Author):

In this manuscript, Fu et al., have demonstrated a solar flow battery (SFB) utilizing a single junction GaAs solar cell. In their demonstration, they report solar-to-electricity conversion efficiency (SOEE) of 13.3%, with a continuous cycling performance of 408 h without much degradation in the performance. Although the results are interesting, it is hard to understand the practical significance of the proposed device. For instance, the authors have used n-type GaAs substrates to fabricate solar cells and claim it to be cheaper compared to p-type GaAs, however the GaAs substrates are much costlier compared to silicon which is widely used in solar cells.

In summary, the results are original and could be considered for a publication after the following issues are addressed.

We sincerely thank the reviewer for his/her positive comments on and approval of our manuscript. With regard to the cost of GaAs solar cells, what we want to convey is that they are much less expensive than tandem III-V solar cells and we have now achieved similar or better SOEE (15.4%) than previously achieved using tandem III-V solar cells (14.1%). The insights gained here lay the foundation to design the strategy to utilize silicon solar cells (which also have high photocurrent densities but lower photovoltages) for efficient SFBs, as alluded to in the discussion part of the manuscript. We have addressed the comments made by the reviewer and revised the manuscript accordingly.

1. The authors have highlighted the usage of single junction solar cell over a multijunction solar cell for coupling with flow battery. However, in practicality any single junction solar cell could be coupled with the flow battery comprising redox couples with redox potential close to voltage output of the solar cell? The authors should explain the rational of choosing a single junction solar cell and coupling with redox couples with lower redox potential which would limit the energy density of the flow battery.

Response: Yes, the analysis developed and the results demonstrated in this work show that, in principle, any single junction solar cell could be coupled with redox flow batteries comprising redox couples with redox potential predicted by our modeling approach (generally close to the maximum output point). However, the choices of single junction vs. tandem solar cells have different strengths and weaknesses, as revealed by this work.

Though tandem junction solar cells with higher photovoltages can enable higher efficiency of the SFBs built with them, this efficiency is gained at the cost of increased complexity and manufacturing cost. To date, the higher price and higher price-to-performance ratio of tandem solar cells have limited their use to niche applications. In the current work, we showed for the first time that a single junction solar cell can still achieve reasonably high SFB efficiency (SOEE) of 15.4% (now further improved from the record 13.3% shown in the original submission). These were achieved by better understanding the voltage matching principles (discussed throughout this manuscript, e.g. Figure 3 and now also Figure 5a) and careful design of the SJ-GaAs solar cells with an unusual “reversed” n-p-n sandwiched layer stacking with cost-effective n-GaAs substrates (discussed in page 4 of the introduction).

Since only single junction solar cells have been commercially deployed at a massive scale, this work opens up the possibility to make practical SFBs with commercially successful solar cells, especially if efficient SFBs can be designed using silicon solar cells (which is indeed a direction we are currently pursuing). However, as will be discussed in detail in the response to the Question 5 below, it is generally more difficult to achieve good voltage match and thus high SPUR and SOEE using single junction solar cells with lower photovoltages, in comparison with tandem solar cells with higher photovoltages. This is precisely the reason why we need more careful analysis and modeling study as shown in this work.

On the other hand, single junction solar cells usually have higher photocurrent densities, which can allow us to charge up higher concentration of electrolytes, which can improve the capacity utilization rate of the electrolytes and the energy density of the SFBs, contrary to what the Reviewer hypothesized (i.e. the lower redox couple potentials could be made up by the higher concentration of the redox couples in the electrolytes). However, to fully take advantage of such higher photocurrent densities, we need to improve the kinetics of the charge transfer on the photoelectrode/electrolyte interface and better engineer the SFB devices, as discussed in the discussion section of the manuscript (around page 18).

2. In page no. 8, should E^0_{Fc} be 0.328 V or 0.382 V in the equation 2?

Response: We sincerely apologize for the typo in the manuscript. It should be 0.382 V in the equation 2. We have now corrected it in the revised manuscript.

3. In page no. 9, the authors mentioned that, higher flow rate of electrolyte causes destruction of the photo electrode. Is this due to the adhesion problem of the protective materials? Please explain the reason for the destruction of the photoelectrode materials at higher flow rate.

Response: We thank the reviewer for the comment. The short answer is Yes. The performance decay at higher flow rate is likely from the physical peeling of the protection layer on the SJ-GaAs photoanode surface, as revealed by the existing Supplementary Fig. 3 (photographs of the photoelectrodes) and Fig. 8 (XPS analysis). We have added more discussion of the X-ray photoelectron spectroscopy (XPS) analysis to discuss these results more clearly for understanding the failure mechanism of the GaAs photoelectrode at higher flow rate in **page 10 of the revised manuscript:**

Post-mortem optical imaging (Supplementary Fig. 3c to 3e) and X-ray photoelectron spectroscopy (XPS) surface analysis (Supplementary Fig. 8) were further performed to understand the failure mechanism of the surface protection layer of Ti/TiO₂/Pt. They suggested that the TiO₂ protection layer (together with the Pt layer on top) disappeared after cycling for the case of the 80 mL min⁻¹ flow rate, but remained mostly intact for the cases of lower flow rates. The disappeared Pt XPS signal and the newly emerged Ga and As signals after cycling at the 80 mL min⁻¹ flow rate indicated that the Ti/TiO₂/Pt protection layer was peeled off during the SFB

cycling test to expose the vulnerable GaAs substrate to electrolyte. Therefore, the observed SFB device performance decay was likely caused by such mechanical damage under the high flow rate. Therefore, in the remaining SFB measurements of this work, the electrolyte flow rate was set to 60 mL min⁻¹ to balance the efficiency and stability concerns.

4. The authors should discuss the energy density of the SFB and make a comparison with the literature?

Response: We thank the reviewer for the suggestions. We agree that the energy density is an important factor for SFB. The current SFB showed energy capacity of 2.41 Ah L⁻¹ (page 9). We had in fact already shown the capacity (mAh L⁻¹) of the SFB (before and after 150 cycles) in Supplementary Fig. 5 and Supplementary Fig. 9c. From this information and the voltage of the SFB, the energy capacity can be estimated to be 1.30 Ah L⁻¹ (the SFB was charged at optimized SOC range from 0 to 54% in this work). The SFB showed a galvanostatic-potentiostatic charge/discharge capacity of 2.41 Ah L⁻¹ and an energy density of 2.05 Wh L⁻¹, which is now mentioned in page 9 and 12 of the manuscript. To further emphasize this point, we have now added the parameter of the capacity in the Figure 6 (the previous Figure 5) which summarizes the SFB performance in representative previous reports by comparing several key parameters: *SOEE* (horizontal axis), current density of the photoelectrode (vertical axis), demonstrated cycling lifetime (the radius of the circles), the pH of the electrolytes is also marked with the color of the data symbol and energy capacity value is noted for each entry. Note that we could not directly compare with energy density of most other reported SFBs because the reported information was not complete.

Updated Fig. 6. SFB performance of representative previous demonstrations in comparison with this work. The number in the circle and the circle radius represent the demonstrated continuous cycling time and their corresponding range, respectively. The fill color of the circle shows the electrolyte pH range. The solar cell structure of each work is also represented by the symbols of triangle (for single junction) and pentagon (for tandem junction). The redox couples and the corresponding energy capacity of the SFB are displayed near each work.

5. In Fig 3 (a), the result suggests that at higher State of Charge (SOCs) of the SFB there is a drastic reduction in the current. Is this reduction arising from the solar cells or due to the redox couples used? In a recent report (*Nature Materials*. 2020 Jul 13:1-6.), the authors demonstrated more than 80% of the current is retained even at 90% SOC.

Response: Thank you for the comment. We agree that the SJ-GaAs SFB had a drastic reduction in current with the BTMAP-Vi/Fc redox couples, which reduced the overall performance of the SFB. The reason for the lower operation current ($I_{operating}$) at higher SOC is illustrated in Figure 3a (reproduced below for convenience): by overlaying the I-V curves of the RFB at the different SOC (1 to 99% SOC) and the I-V curve of SJ-GaAs photoanode at 50% SOC, the operation points of the SFB can be found at the intersection points. The $I_{operating}$ was very sensitive to the *LSV* behavior of the photoelectrode at different SOC.

Fig. 3. a) Estimation of $SOEE_{ins}$ for SFB built with SJ-GaAs photoanode and BTMAP redox couples.

In comparison with our recently published *Nature Materials* paper, the voltage match is not yet as optimal for the SJ-GaAs SFB shown here as demonstrated for the tandem perovskite/silicon solar cells in *Nature Materials* paper, where a nearly optimal voltage match was achieved. In fact, according to the simulation, the E^0_{cell} of the BTMAP-Vi/Fc redox couples was actually too high for the actual GaAs photoanodes, even though it was optimal based on the solid-state GaAs solar cells. We have recently studied a new SJ-GaAs SFB by using the better matched of N^{Mc} -TEMPO and BTMAP-Fc redox couples that could deliver the E^0_{cell} of 0.558 V and demonstrated further

improved SOEE for this SFB to 15.4%. These results are shown in the new Figure 5 in the revised manuscript (reproduced below).

New Fig. 5. The simulated $SOEE - E_{cell}^0$ curve, CV, and cycling performance of a SFB device with SJ-GaAs photoanode and BTMAP-Fc/ N^{Me} -TEMPO redox couples. **a** The numerically calculated $SOEE$ as a function of E_{cell}^0 (red curve) by using the LSV data from the SJ-GaAs photoelectrode (the data from Figure. 3a) and solid-state SJ-GaAs cell (blue curve, the same as Fig. 1d). **b** The cyclic voltammograms of 5.0 mM BTMAP-Fc and 5.0 mM N^{Me} -TEMPO collected at a scanned rate of $10\ mV\ s^{-1}$ on a glassy carbon electrode in 1.0 M NaCl supporting electrolyte. **c** Cell potential and photocurrent density of the intergrade SFB device during cycling. **d** CE (blue triangles), VE (yellow triangles) and $SOEE$ (red circles) of the integrated SFB device over 10 cycles. The SFB cycling was performed with 0.1 M BTMAP-Fc/ N^{Me} -TEMPO redox couples in catholyte/anolyte and a flow rate of $60\ mL\ min^{-1}$ over 10 cycles under one Sun solar illumination. Each cycle started with 36 min of bias-free solar charging process followed by a galvanostatic discharging step at $-11\ mA$ until reaching the cutoff potential (0.25 V).

But this brings a broader and general discussion about the comparison between single junction solar cells with lower V_{oc} and tandem solar cells with higher V_{oc} : Because the $SOEE$ of the SFB is very sensitive to the LSV behavior of the photoelectrode, the decreased FF of the GaAs photoanode would significantly alter its voltage matching with the redox couples in $SFBs$ (see the red curve in Supplementary Fig. 11a, reproduced below). In contrast, the same SOC swing will not create as much voltage mismatch in higher voltage cells such as tandem III-V cell (see

the orange curve in Supplementary Fig. 11a), because of the smaller *relative* voltage shift through the J - V curve. Accordingly, the instantaneous SOEE is much less sensitive to the SOC for a higher V_{oc} cell (Supplementary Fig. 11b), therefore tandem solar cells with higher V_{oc} are more likely to achieve better voltage match and higher $SPUR$ of the SFB. This discussion is now added to page 15 of the revised manuscript and this new Supplementary Fig. 11 and the associated discussion was added in Supplementary Information.

New Supplementary Figure 11. Overlaid hypothetical current-potential behaviors between RFBs and two different photoelectrodes with different V_{oc} . **a** J - V performance of the SJ-GaAs (red) and triple junction III-V (orange, replotted from the data reported in Ref.15) photoanode measured at solar cell mode overlaid with the I-V curves of RFBs at different SOC simulated with the respective optimally matched E°_{cell} of 0.59 V (green lines) and 1.72 V (blue lines). The SJ-GaAs photoelectrode were measured at the flow rate of 60 mL min⁻¹ under one Sun illumination at 50% SOC. **b** Normalized $SOEE_{ins}$ as a function of SOC for both solar cells.

6. In page no 17, was n-GaAs substrate diced into 100 mm × 100 mm square pieces or 10 mm x 10 mm square pieces?

Response: We apologize for mistake. It should be 10 mm x 10 mm square pieces. We have corrected this mistake.

Reviewer #3 (Remarks to the Author):

Solar flow battery has become a rapid trend for the solar cell and flow battery communities. Given the poor chemical resistance of solar cells, the flow chemistry in solar flow battery is oftentimes non-corrosive and mild neutral. Because of this limit, the concentration of active materials is always very low, which is actually at least three orders of magnitude lower than that of commercial vanadium flow battery. This has caused lots of energy waste for the pumping system. For instance, given a flow rate of 50 ml/min, the energy stored by SFB is 1/1000 of that by traditional FB that is charged by traditional solar cell. In this sense, the technological significance of SFM is rather poor. Having that said, it is still of great value to explore the novel integrated photo-electrochemistry at the interface. Finding new chemistry will be always valued for fundamental acquisition of new knowledge. In this work, the authors adopted existing systems of redox couples and GaAs photo absorber. It is important to be aware of the toxicity of As which is not in fact popular in solar cell field. This work presents the innovative engineering by integrating two parts from existing knowledge and technical research database - it is for sure remarkable optimization experiments were conducted to achieve the reported high performance. Yet scientifically, very limited new knowledge can be obtained from this work.

Response:

We thank the reviewer for the overall positive assessment of the manuscript, especially for the comment on the contributions and great value of this work to the rapidly developing field and the “innovative engineering” to “achieve the reported high performance”. We address the few general criticisms in the comments below:

“the concentration of active materials is always very low”, thus “energy waste for the pumping system”. We appreciate this insightful comment and the perspective from the whole system level. First, here we intentionally used the emerging organic redox flow batteries, which are still on the verge of commercialization, so that the overall engineering of these new RFB systems lags behind the commercially deployed vanadium RFBs that have been developed for over 3 decades. It is expected (hoped) that organic RFBs will be further improved in terms of the concentration of redox couples, capacity, energy density, cycling lifetime, and eventually become less expensive than vanadium RFBs due to the intrinsic lower cost of organic materials than vanadium. Second, in terms of increasing the concentration of active redox couple materials in SFBs, this is in fact a direction this work is helping to improve on. As discussed in page 14-18, we are pushing the boundary on designing efficient SFBs using single junction solar cells with higher photocurrent densities. *Higher photocurrent densities can allow us to charge up higher concentrations of electrolytes, which can improve the capacity utilization rate of the redox active materials and the energy density of the SFBs.* However, to fully take advantage of such higher photocurrent densities, we need to improve the kinetics of the charge transfer on the photoelectrode/electrolyte interface and better engineer the SFB devices, as discussed in the discussion section of the manuscript.

We agree with the comment about the toxic As in GaAs solar cells. In fact, we would also admit that GaAs solar cells are generally still considered too expensive for practical solar cells. But we would like to put the progress here into perspectives, single junction GaAs solar cells are much less expensive than tandem III-V solar cells previously utilized, and we have now achieved even better SOEE (15.4%) than previously achieved using tandem III-V solar cells (14.1%). More importantly, *the insights gained here lay the foundation to design the strategy to utilize silicon solar cells (which also have high photocurrent densities but lower photovoltages) for efficient SFBs*, as alluded to in the discussion part of the manuscript.

Scientifically, *the new design strategies for utilizing single junction solar cells for more efficient SFBs and the detailed modeling analysis demonstrated here are the new understanding* we introduce in this work, and we believe that they have general significance beyond the specific “engineering” demonstration using the GaAs solar cells and the specific redox couples. As discussed in detail in the response to Question #1 by Reviewer #1, there can be several potential benefits of integrated SFBs over separated PV+ RFBs in terms of system cost, efficiency and device thermal management. To realize these potential benefits, significant developments are always needed to improve any nascent technologies to make them competitive with established ones: it is very important to design and fabricate efficient photovoltaic (PV) cells and photoelectrodes with suitable protection/modifications while developing the suitable robust redox couples with optimal potentials, that addresses the issues of cost, cycling stability, voltage match, and slow carrier transfer kinetics across the photoelectrode/electrolytes interface so we further enhance the performance and stability of SFBs. These are what we are trying to make progress on in this work.

REVIEWERS' COMMENTS

Reviewer #1 (Remarks to the Author):

The revised manuscript has addressed all the comments of the reviewers, and made considerable changes to further improve the paper quality. I don't have further comments, and would recommend the acceptance of the manuscript.

Reviewer #2 (Remarks to the Author):

The authors have carefully addressed my comments and I appreciate them for this. This article can be published now.